# INSIGHTMAPPER: A CLOSER LOOK AT INNER-INSTANCE INFORMATION FOR VECTORIZED HIGH-DEFINITION MAPPING

## ABSTRACT

Vectorized high-definition (HD) maps contain detailed information about surrounding road elements, which are crucial for various downstream tasks in modern autonomous vehicles, such as motion planning and vehicle control. Recent works have attempted to directly detect the vectorized HD map as a point set prediction task, resulting in significant improvements in detection performance. However, these methods fail to analyze and exploit the inner-instance correlations between predicted points, impeding further advancements. To address this issue, we investigate the utilization of inner-**INS**tance information for vectorized h**IGH**-definition mapping through **T**ransformers and introduce InsightMapper. This paper presents three novel designs within InsightMapper that leverage inner-instance information in distinct ways, including hybrid query generation, inner-instance query fusion, and inner-instance feature aggregation. Comparative experiments are conducted on the NuScenes dataset, showcasing the superiority of our proposed method. InsightMapper surpasses previous state-of-the-art (SOTA) methods by 5.78 mAP and 7.03 TOPO, which assess topology correctness. Simultaneously, InsightMapper maintains high efficiency during both training and inference phases, resulting in remarkable comprehensive performance.

## 1 INTRODUCTION

High-definition maps (HD maps) play a critical role in today's autonomous vehicles (Gao et al., 2020; Liang et al., 2020; Da & Zhang, 2022), as they contain detailed information about the road, including the position of road elements (e.g., road boundaries, lane splits, pedestrian crossings, and lane centerlines), connectivity, and topology of the road. Without the assistance of HD maps for perceiving and understanding road elements, unexpected vehicle behaviors may be encountered, such as incorrect path planning results or even vehicle collisions.

Typically, HD maps are created by offline human annotation, which is labor-intensive, inefficient, and expensive. Although there are works proposing to make such an offline process automatic (Xu et al., 2022b; Xie et al., 2023; Xu et al., 2023), it is not possible to recreate and update the HD map frequently when the road network has been modified, such as when a new road is built or an existing road is removed. Unlike these previous works, this paper studies HD map detection in an online manner based on vehicle-mounted sensors (e.g., cameras, LiDARs).

Early work considers road element mapping as a semantic segmentation task in birds'-eye view (BEV) (Philion & Fidler, 2020; Ng et al., 2020; Li et al., 2022b), in which road elements are predicted in raster format (i.e., pixel-level semantic segmentation mask). However, rasterized road maps cannot be effectively utilized by downstream tasks of autonomous vehicles, such as motion planning and vehicle control (Gao et al., 2020; Liang et al., 2020; Liu et al., 2021). Moreover, it is challenging to distinguish instances from the rasterized map, especially when some road elements overlap with each other or when complicated topology is encountered (e.g., road split, road merge, or road intersections). To alleviate these problems, HDMapNet (Li et al., 2021) proposes hand-crafted post-processing algorithms to better obtain road element instances for vectorized HD maps. However, HDMapNet still heavily relies on rasterized prediction results, which restricts

it from handling topologies and complicated urban scenes. Recently, some work has resorted to predicting vectorized HD maps directly (Can et al., 2021a; Liu et al., 2022; Liao et al., 2022; Shin et al., 2023). A two-stage hierarchical set prediction method is proposed in VectorMapNet (Liu et al., 2022). After predicting key points of road elements in the HD map, VectorMapNet sequentially generates intermediate points. Although VectorMapNet is considered the first online vector map detection work, the sequential operation degrades its efficiency and model performance. MapTR (Liao et al., 2022) further proposes to use DETR (Carion et al., 2020; Zhu et al., 2020) for vectorized HD map detection as a point set prediction problem. The output of MapTR is a set of points, which are then grouped into road element instances. Even though MapTR achieves state-of-the-art (SOTA) performance on vectorized HD map detection so far, it fails to understand and utilize the correlation between points to further enhance model performance.

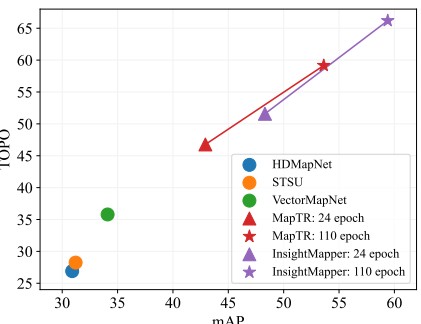

Figure 1: Comparison of vectorized HD map detection methods. All methods are evaluated on the NuScenes *val* set. The y-axis displays topology level correctness as per (He et al., 2018), and the x-axis presents the mAP results. InsightMapper outperforms other baseline models in both metrics.

In conventional DETR object detection tasks, objects can be assumed to be independent and identically distributed (i.i.d.). However, in the vectorized map detection task, there exists a strong correlation between predicted points, particularly points within the same instance, known as *inner-instance* points. Due to the failure to utilize such correlation information between points, MapTR does not achieve further improvements in its final performance. Therefore, the method to better make use of inner-instance point information is carefully studied in this paper.

In this work, we study the inner-**INS**tance information for vectorized h**IGH**-definition mapping by **T**ransformer, and propose a novel model named InsightMapper, which can utilize inner-instance point information for improved online HD map detection. First, the hierarchical object query generation method in MapTR is refined. The incorrect inter-instance correlation is removed and only inner-instance correlation is maintained. Then, a query fusion module is added before the transformer decoder, which fuses object queries to make the prediction more consistent. Finally, a novel masked inner-instance self-attention module is incorporated into transformer decoder layers, in order to better aggregate features of inner-instance points. All methods are evaluated on the NuScenes *validation* set. Compared with the previous SOTA method MapTR, InsightMapper achieves 5.78% higher mAP as well as 7.03% topological correctness. Meanwhile, the efficiency of both the training and inference phases is well maintained. Comparison results are visualized in Figure 1. The contributions of this work can be summarized below:

- We investigate the correlation between inner-instance points, demonstrating that utilizing inner-instance point information can effectively enhance the final performance.

- To better leverage inner-instance information, we introduce a new model called InsightMapper for online HD map detection. InsightMapper incorporates three novel modules with distinct functions, including query generation, query fusion, and inner-instance self-attention.

- We evaluate all module designs and baselines on the NuScenes *validation* set. InsightMapper outperforms all baseline models and maintains competitive efficiency.

## 2 RELATED WORKS

**Road Element Detection.** The online HD map detection task aims to predict the graph of road elements with vehicle-mounted sensors, drawing inspiration from similar tasks such as road-network detection (Bastani et al., 2018; He et al., 2020; Xu et al., 2022a; 2023), road-curb detection in aerial images (Xu et al., 2021), and road lane line detection (Homayounfar et al., 2018; 2019). DagMapper (Homayounfar et al., 2019) seeks to detect vectorized lane lines from pre-built point cloud maps using iterations. However, DagMapper only considers simple topology changes of lane lines and struggles to handle complex road intersections. RNGDet++ (Xu et al., 2023) applies DETR along

with an imitation learning algorithm (Ross et al., 2011) for road-network detection, achieving state-of-the-art performance on multiple datasets. Although these methods demonstrate satisfactory graph detection accuracy, they suffer from poor efficiency due to their iterative algorithms. Given the strict online requirements of autonomous vehicles, they are not suitable for online HD map detection.

**HD Map Detection.**   The road element detection task was initially a subtask of BEV detection (Philion & Fidler, 2020; Ng et al., 2020; Li et al., 2022b; Liu et al., 2023; Li et al., 2023b). Recently, given the importance of HD maps, several works have focused on directly optimizing HD map detection (Can et al., 2021b; Li et al., 2021; Mi et al., 2021; Liu et al., 2022; Liao et al., 2022; Xu et al., 2022b; Xie et al., 2023; Shin et al., 2023; Qiao et al., 2023). However, most of these works either involve offline HD map creation (Xu et al., 2022b; He & Balakrishnan, 2022; Xie et al., 2023) or only detect one specific road element (Can et al., 2021b;a; Qiao et al., 2023). HDMapNet (Li et al., 2021) is considered the first work specifically designed for multiple road element detection (i.e., road boundary, lane split and pedestrian crossing). However, HDMapNet outputs rasterized results, requiring complicated hand-crafted post-processing to obtain vectorized HD maps. To address this issue, VectorMapNet (Liu et al., 2022) is believed to be the first work detecting vectorized HD maps on-the-fly. However, it consists of two stages, and its efficiency is significantly impacted by sequential operations. In contrast, MapTR (Liao et al., 2022) employs deformable DETR (Zhu et al., 2020) to design an end-to-end vector map detection model, which greatly simplifies the pipeline and delivers better detection performance. However, it does not investigate the correlation between points, limiting further improvements.

**Detection by Transformer.**   DETR (Carion et al., 2020) is the first end-to-end transformer-based object detection framework. Compared with previous CNN-based methods (Girshick, 2015; He et al., 2017), DETR eliminates the need for anchor proposals and non-maximum suppression (NMS), making it simpler and more effective. To address the issue of slow convergence, subsequent works propose accelerating DETR training through deformable attention (Zhu et al., 2020), denoising (Li et al., 2022a), and dynamic query generation (Wang et al., 2022; Zhang et al., 2022; Li et al., 2023a). Intuitively, most of these DETR refinements could be adapted to the HD map detection task since our proposed InsightMapper relies on DETR. However, unlike conventional object detection tasks where objects are approximately independent and identically distributed (i.i.d.), the detected points in HD maps exhibit strong correlations, particularly among inner-instance points. This inherent difference renders some refined DETR methods inapplicable to our task.

## 3   POINT CORRELATION

### 3.1   PRE-PROCESSING: VECTOR MAP DECOMPOSITION AND SAMPLING

Let $G$ denote the original vector map label of the scene, which consists of vertices $V$ and edges $E$. The vector map contains multiple classes of road elements, including pedestrian crossings, road dividers, road boundaries, and lane centerlines. Among them, the first three classes of road elements are simple polylines or polygons without intersection points. While lane centerlines have more complicated topology, such as lane split, lane merge, and lane intersection. To unify all vector elements, the vector map is decomposed into simpler shapes (i.e., polylines and polygons) without intersections. Any vertices in the vector map with degrees larger than two (i.e., intersection vertices) are removed from $G$, and incident edges are disconnected. In this way, a set of simple polylines and polygons without intersections is obtained as $G^* = \{l_i\}_{i=1}^{N^*}$, where $G^*$ is an undirected graph. Each shape $l_i$ is defined as an instance, and $N^*$ denotes the overall number of instances in a vector map.

Following MapTR, to enhance the parallelization capacity of the model, each instance is evenly re-sampled with fixed-length points as $l_i = (v_1, ..., v_j, ...v_{n_p})$. $l_i$ is ordered from $v_1$ to $v_{n_p}$, where $n_p$ is the number of sampled points for each instance. For polygon instances, $v_1$ is equal to $v_{n_p}$. The pre-processing module is visualized in Figure 2.

### 3.2   INNER-INSTANCE AND INTER-INSTANCE CORRELATION

Unlike conventional object detection tasks, where objects can be approximated as independent and identically distributed (i.i.d.), strong correlations exist between predicted points in the vectorized

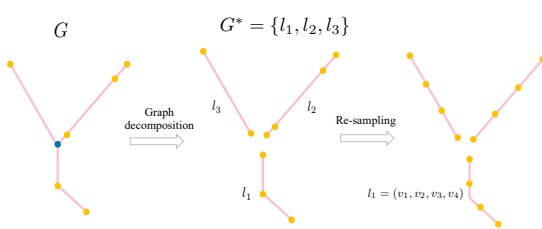

Figure 2: Pre-processing of the vector map: Pink lines represent edges, orange points indicate vertices, and the blue point is the intersection vertex with a degree larger than two. The intersection is removed to simplify the graph, and each obtained instance is then evenly re-sampled into $n_p$ vertices ($n_p = 4$ in this example).

Figure 3: Visualization of inner- and inter-correlations. Green lines represent the inner-instance correlation between the blue point and other points within the same instance, while red lines depict the inter-instance correlation, which should be blocked to prevent interference.

HD map detection task. These correlations can be classified into inner-instance correlation and inter-instance correlation, as visualized in Figure 3.

Inner-instance correlation is crucial for point coordinates prediction. Points within the same instance can collaborate by exchanging inner-instance information to produce smoother and more precise predictions. Without this collaboration, points of the same instance may produce independent predictions, leading to zig-zag or incorrect instances. In another word, inner-instance correlation can serve as additional constraints to refine the predicted vector map. Conversely, inter-instance correlations occur between a point and points belonging to other instances. Usually, inter-instance correlation distracts the inner-instance information exchange, degrading the final performance. Thus, inter-instance information should be blocked.

In the previous state-of-the-art (SOTA) method, MapTR, the correlation of points is not correctly analyzed and leveraged: inner-instance information is not fully utilized, and incorrect inter-instance information is introduced, limiting further improvement. In subsequent sections, we propose InsightMapper to better handle and leverage point correlations, aiming to achieve enhanced vectorized map detection performance.

# 4 METHODOLOGY

## 4.1 SYSTEM OVERVIEW

Building on MapTR (Liao et al., 2022), our proposed InsightMapper is an end-to-end trainable network for online vectorized HD map detection. Unlike MapTR, which treats queries as independently distributed, InsightMapper utilizes inner-instance information to fuse object queries and aggregate intermediate features in deformable transformer decoder layers. Otherwise, adjacent vertices may produce independent position regression results, leading to zigzag or even incorrect instances. With the assistance of inner-instance information, the model can better optimize and refine the positions of points within an instance, significantly enhancing the final performance.

To better leverage inner-instance information and further enhance the final detection performance, we introduce three main designs in InsightMapper: (1) Hybrid query generation: We propose a hybrid query generation method to replace the hierarchical query in MapTR, preventing inter-instance interference and preserving inner-instance information. (2) Inner-instance query fusion: After generating queries, we propose to fuse queries within the same instance based on inner-instance features. (3) Inner-instance feature aggregation: A new masked-self-attention layer is added to deformable decoder layers to further aggregate features of points within the same instance. More details of these modules are discussed in subsequent sections.

**Network structure.** InsightMapper is an end-to-end trainable encoder-decoder transformer model. Following BEVformer (Li et al., 2022b), InsightMapper first projects perspective view camera images into the bird's-eye-view (BEV). After obtaining the BEV features, InsightMapper employs deformable attention layers (Zhu et al., 2020) to process input decoder object queries. Each

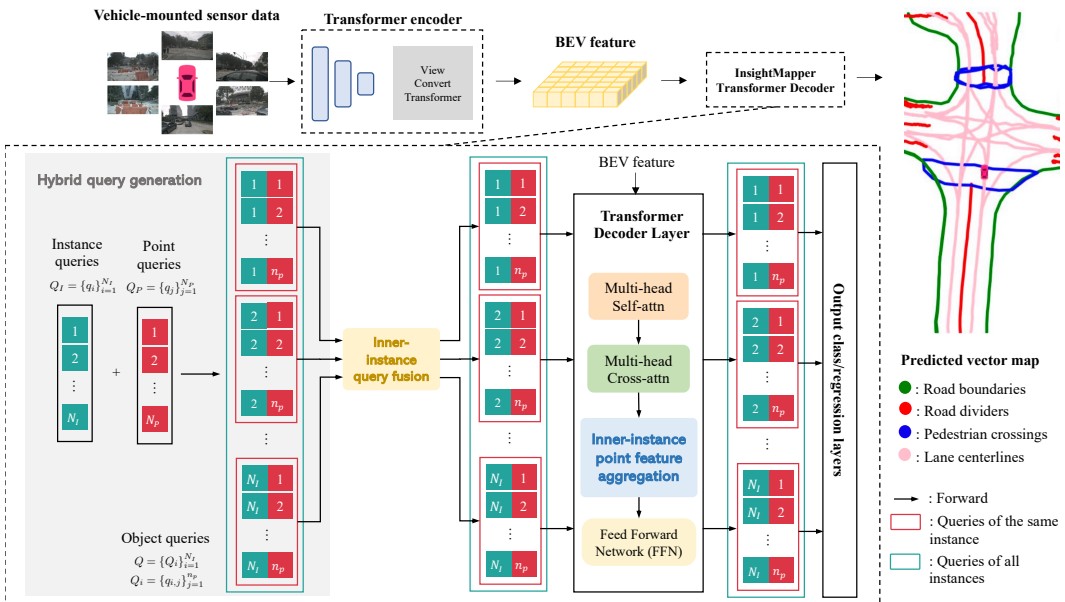

Figure 4: System overview. InsightMapper is an end-to-end trainable transformer model with an encoder-decoder structure. The transformer encoder projects perspective-view camera images into a bird's-eye view (BEV). Subsequently, the transformer decoder detects vector instances by predicting point sets. To better utilize inner-instance information, we: (a) propose a hybrid query generation scheme; (b) add a module to fuse input queries based on the inner-instance information; (c) insert an inner-instance feature aggregation module into transformer decoder layers for point feature exchange within an instance.

object query predicts one point, including the object class and the regression point position. The overall structure of InsightMapper is visualized in Figure 4.

## 4.2 QUERY GENERATION

Unlike conventional object detection tasks, where queries are independent and identically distributed (i.i.d.), the points to be detected in our task exhibit strong correlations. There are several methods to generate object queries as input for the transformer decoder, including the naive scheme, hierarchical scheme, and hybrid scheme. All query generation schemes are visualized in Figure 5.

**Naive query generation.** This method assumes points are i.i.d. and generates queries randomly, without utilizing inner-instance information. Let $N_I$ denote the number of predicted instances, which is clearly larger than $N^*$. Since each instance contains $n_p$ points, the naive object queries consist of $N_I \cdot n_p$ randomly generated i.i.d. queries. This query generation method tends to exhibit degraded performance due to the insufficient incorporation of inner-instance constraints.

**Hierarchical query generation.** To address the aforementioned issue of the naive scheme, a hierarchical query generation scheme is proposed in MapTR (Liao et al., 2022). Let $Q_{ins} = \{q_i^I\}_{i=1}^{N_I}$ denote instance queries and $Q_{pts} = \{q_j^P\}_{j=1}^{n_p}$ denote point queries. The object queries for vector map detection are then the pairwise addition of $Q_{ins}$ and $Q_{pts}$, as shown below:

$$Q = \{q_{i,j} = q_i^I + q_j^P | q_i^I \in Q_{ins}, q_j^P \in Q_{pts}\} \tag{1}$$

where $|Q| = |Q_{ins}| \cdot |Q_{pts}| = N_I \cdot n_p$. For points within the same instance $\{q_{i,j}\}_{j=1}^{n_p} = \{q_i^I + q_j^P\}_{j=1}^{n_p}$, they share the same learnable instance embedding $q_i^I$, which can be treated as a means to exchange inner-instance information. By using $q_i^I$ as a bridge, object queries within the same instance can better collaborate with each other for point prediction.

Although this hierarchical query generation scheme exhibits improved performance compared to the naive one, it introduces unexpected inter-instance interference. Intuitively, queries between different instances should not be correlated (i.e., inter-instance queries can be treated as independent). However, the $j$-th query of different instances (i.e., $\{q_{i,j}\}_{i=1}^{N_I} = \{q_i^I + q_j^P\}_{i=1}^{N_I}$) share the same point query $q_j^P$, indicating that this scheme creates information exchange between the $j$-th point of all instances. This incorrect inter-instance information exchange hinders the model from achieving further improvement.

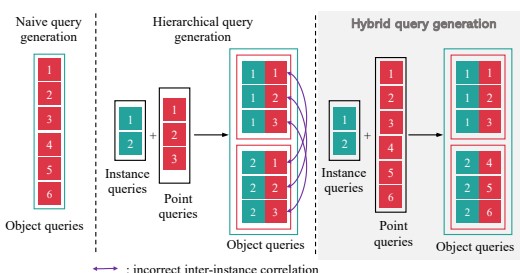

Figure 5: Query generation schemes. For concise visualization, the number of instances $N_I$ is 2, and the number of points per instance $n_p$ is 3. The hybrid scheme can effectively facilitate inner-instance information exchange and prevent incorrect inter-instance interference (purple lines) simultaneously.

**Hybrid query generation.**  The naive scheme does not take into account any information exchange between queries, whereas the hierarchical one introduces detrimental inter-instance correlations. To address these issues, this paper presents a hybrid query generation method that mitigates the drawbacks of the aforementioned schemes while maintaining appropriate inner-instance information exchange.

Let the instance queries be $Q_{ins} = \{q_i^I\}_{i=1}^{N_I}$, and the point queries be $Q_{pts} = \{q_j^P\}_{j=1}^{N_P} = \{q_j^P\}_{j=1}^{N_I \cdot n_p}$. The point queries are divided into $N_I$ instance groups, and a point query $q_j^P$ is assigned to the $\lceil \frac{j}{n_p} \rceil$-th instance. Consequently, the final object query is the sum of a point query $p_j^P$ and its assigned instance query $p_k^I$, where $k = \lceil \frac{j}{n_p} \rceil$. The object query set can be expressed as:

$$Q = \{q_k^I + q_j^P | q_j^P \in Q_{pts}\} \tag{2}$$

where $|Q| = |Q_{pts}| = N_P = N_I \cdot n_p$. In contrast to the hierarchical scheme, each point query in the hybrid scheme is utilized only once to generate the object query, thereby preventing unintended inter-instance information exchange. Simultaneously, point queries belonging to the same instance are summed with a shared instance query, establishing the inner-instance connection. The hybrid query generation method can be considered a combination of the naive and hierarchical schemes, effectively mitigating their respective drawbacks.

### 4.3 Inner-instance query fusion

The input object queries for the transformer decoder are generated from instance queries and point queries. Although the generated queries can leverage some inner-instance information, this information exchange is indirect and inflexible. The instance query is distributed equally among all inner-instance points, while a more precise point-to-point information exchange cannot be realized. As a result, a query fusion module is introduced to further utilize inner-instance information.

Let $Q_i = \{q_{i,j}\}_{j=1}^{n_p}$ represent the set of object queries belonging to the $i$-th instance, and $q_{i,j}$ denote the $j$-th point of the $i$-th instance. $q_{i,j}$ is correlated with all other queries in $Q_i$. To better fuse inner-instance object queries, each query is updated by a weighted sum of all queries in $Q_i$ as:

$$q_{i,j} = f(q_{i,j}, Q_i) = \sum_{k=1}^{n_p} w_{i,j,k} \phi(q_{i,k}) \tag{3}$$

where $w_{i,j,k}$ demonstrates weights for query fusion.  $f(\cdot)$ is the fusion function and $\phi(\cdot)$ is the kernel in case nonlinear transformation is needed. $f(\cdot)$ could be realized by handcraft weights, fully connected layers, or self attention.

In conventional object detection tasks, object queries are assumed to be independent and identically distributed (i.i.d.), making query fusion unnecessary. However, in the task of vector map detection, the query fusion module effectively aligns the update of queries and enables each point to pay more

attention to neighboring points. Without the query fusion module, queries within the same instance cannot be aware of each other, rendering them "blind". The lack of information on neighboring queries prevents them from forming precise shapes, leading to a degradation in final performance.

## 4.4 INNER-INSTANCE FEATURE AGGREGATION

In addition to object query manipulation, InsightMapper performs inner-instance feature aggregation in the transformer decoder layers by incorporating an additional masked inner-instance self-attention module. Inspired by (He et al., 2022), inner-instance attentions are randomly blocked with probability $\epsilon$ for robustness. The decoder layer of InsightMapper is depicted in Figure 6.

The inner-instance self-attention module resembles the original self-attention module but features a specially designed attention mask. As illustrated in Figure 6, the attention mask of inner-instance points is set to zero (colored grids), indicating that attention between points within the same instance is allowed. Conversely, for points belonging to different instances (i.e., inter-instance points), the corresponding attention mask values are set to one, blocking attention between them. This method encourages the model to focus more on inner-instance information, resulting in more consistent predictions. To further enhance the robustness of this module, the inner-instance attention (colored grids) has an $\epsilon$ probability of being blocked.

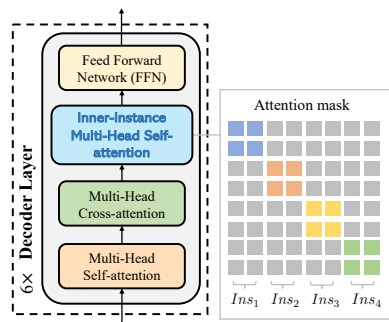

Figure 6: Decoder of InsightMapper. A masked inner-instance self-attention module is incorporated into decoder layers. In this module, the attention between points belonging to different instances is blocked (grey grids). Only inner-instance attention is allowed (colored grids). Colored grids are randomly blocked with $\epsilon$ probability (set to grey) for robustness.

An alternative method involves placing the proposed attention module before the cross-attention module. However, self-attention before cross-attention should treat all queries equally to prevent duplicated predictions. Consequently, implementing inner-instance self-attention before the cross-attention module leads to a degradation in the final performance. Thus inner-instance feature aggregation should be placed after the cross-attention layer. More discussion about the decoder network structure is provided in Appendix C.2.

## 5 EXPERIMENTS

**Datasets.** In this paper, we conduct all experiments on the NuScenes dataset (Caesar et al., 2020) and the Argoverse-2 dataset (Wilson et al., 2023). These two datasets comprise hundreds of data sequences captured in various autonomous driving scenarios, encompassing diverse weather conditions and time periods. In accordance with previous studies (Li et al., 2021; Can et al., 2021a; Liao et al., 2022; Liu et al., 2022), the detected vectorized HD map should encompass four types of road elements: road boundary, lane split, pedestrian crossing, and lane centerline. The perception range for the X-axis and Y-axis in the BEV is set to $[-15m, 15m]$ and $[-30m, 30m]$, respectively.

**Implementation details.** In this paper, we perform experiments on a machine equipped with 8 RTX-3090 GPUs. During the training phase, all GPUs are utilized, whereas only a single GPU is employed for inference. To ensure a fair comparison, different methods are trained using their respective optimal settings as reported in previous literature. Specifically, for our proposed InsightMapper, we adopt multiple settings akin to the previous SOTA method, MapTR. In the comparative experiments, InsightMapper is accessed with ResNet (He et al., 2016) and VoVNetV2-99 (Lee & Park, 2020) as the backbone. All ablation studies are conducted for 24 epochs, employing ResNet50 as the backbone network.

**Evaluation metrics.** All models are assessed using two types of metrics. The first metric, average precision (AP), gauges the instance-level detection performance, employing Chamfer distance for matching predictions with ground truth labels. To ensure a fair comparison, we follow previous works to calculate multiple $AP_\tau$ values with $\tau \in \{0.5, 1.0, 1.5\}$, and report the average AP. To

Table 1: Quantitative results of comparison experiments on the NuScenes *validation* set. Colored numbers show differences between InsightMapper and the SOTA baseline under the same experiment setting. "-" represents that the result is not available. "V2-99" and "Sec" correspond to VoVNetV2-99 (Lee & Park, 2020) and SECOND (Yan et al., 2018).

| Methods | Epochs | Backbone | Modality | $AP_{ped}$ | $AP_{div}$ | $AP_{bound}$ | $AP_{center}$ | mAP | TOPO |
|---|---|---|---|---|---|---|---|---|---|
| HDMapNet | 30 | Effi-B0 | C | 4.41 | 23.73 | 58.17 | 37.29 | 30.90 | 29.79 |
| STSU | 110 | R50 | C | - | - | - | 31.21 | 31.21 | 32.11 |
| VectorMapNet | 130 | R50 | C | 28.66 | 39.74 | 33.06 | 34.93 | 34.10 | 39.90 |
| MapTR | 24 | R18 | C | 24.14 | 37.60 | 41.11 | 29.98 | 33.21 | 38.47 |
| MapTR | 24 | R50 | C | 37.92 | 46.25 | 50.07 | 37.47 | 42.93 | 46.77 |
| MapTR | 24 | V2-99 | C | 44.57 | 56.25 | 60.58 | 46.30 | 51.92 | 55.16 |
| MapTR | 24 | R50&Sec | C&L | 49.47 | 58.98 | 66.72 | 44.89 | 55.01 | 56.77 |
| MapTR | 110 | R50 | C | 49.16 | 59.12 | 58.93 | 47.26 | 53.62 | 59.16 |
| InsightMapper | 24 | R18 | C | 31.14(↑7.00) | 43.25(↑5.65) | 42.36(↑1.25) | 34.23(↑4.25) | 37.74(↑4.53) | 42.53(↑4.06) |
| InsightMapper | 24 | R50 | C | 44.36(↑6.44) | 53.36(↑7.11) | 52.77(↑2.70) | 42.35(↑4.88) | 48.31(↑5.38) | 51.58(↑4.81) |
| InsightMapper | 24 | V2-99 | C | 51.16(↑6.59) | 63.71(↑7.46) | 64.47(↑3.89) | 51.40(↑5.10) | 57.68(↑5.76) | 60.00(↑4.84) |
| InsightMapper | 24 | R50&Sec | C&L | **56.00**(↑6.53) | 63.42(↑4.44) | **71.61**(↑4.89) | 52.85(↑7.96) | **60.97**(↑5.96) | 62.51(↑5.74) |
| InsightMapper | 110 | R50 | C | 55.42(↑6.26) | **63.87**(↑4.75) | 65.80(↑6.87) | **54.20**(↑6.94) | 59.40(↑5.78) | **66.19**(↑7.03) |

Table 2: Quantitative results of comparison experiments on the Argoverse 2 *validation* set. Colored numbers show differences between InsightMapper and the SOTA baseline under the same experiment setting.

| Methods | Epochs | Backbone | Modality | $AP_{ped}$ | $AP_{div}$ | $AP_{bound}$ | $AP_{center}$ | mAP | TOPO |
|---|---|---|---|---|---|---|---|---|---|
| MapTR | 6 | R50 | C | 52.88 | 63.68 | 61.18 | 59.73 | 59.37 | 75.79 |
| InsightMapper | 6 | R50 | C | **55.61**(↑2.73) | **66.60**(↑2.92) | **62.58**(↑1.40) | **62.67**(↑2.94) | **61.87**(↑2.50) | **77.58**(↑1.79) |

Table 3: Quantitative results of comparison experiments on the NuScenes *validation* set without centerline.

| Methods | Epochs | Backbone | Modality | $AP_{ped}$ | $AP_{div}$ | $AP_{bound}$ | mAP |
|---|---|---|---|---|---|---|---|
| MapTR | 24 | R50 | C | 46.04 | 51.58 | 53.08 | 50.23 |
| InsightMapper | 24 | R50 | C | **48.44**(↑2.40) | **54.68**(↑3.10) | **56.92**(↑3.84) | **53.35**(↑3.12) |

measure the topology correctness of the results, the topological metric TOPO that is widely used in past works (He et al., 2018; 2020; He & Balakrishnan, 2022; Xu et al., 2023) is reported. For all metrics, a larger value indicates better performance.

## 5.1 COMPARISON EXPERIMENTS

In this section, we compare InsightMapper with previous SOTA methods using the aforementioned evaluation metrics. Quantitative comparison results are reported in Table 1. Additionally, efficiency comparisons are presented in Table 4. Qualitative visualizations are visualized in Figure 7. From Table 1, it is evident that our proposed InsightMapper attains the best evaluation performance across all metrics while maintaining competitive efficiency. Compared to the previous SOTA method, MapTR, InsightMapper improves the AP of all road elements by approximately 5%, and the mAP increases by 5.38% for 24 epoch experiments and 5.78% for 110 epoch experiments. The topological metric also sees an improvement of more than 5%. Table 3 presents the outcomes without centerlines. Excluding centerlines simplifies the detection task and scenes, resulting in a slight decrease in InsightMapper's performance enhancement. Nonetheless, it still attains an improvement of around 3 mAP. The evaluation outcomes on the Argoverse-2 dataset (Wilson et al., 2023) can be found in Table 2. Argoverse-2 comprises simpler scenarios compared to Nuscenes, resulting in enhanced baseline performance and marginally smaller improvements. Despite delivering superior detection results, Table 4 demonstrates that InsightMapper maintains high efficiency, with negligible impact on training and inference costs. Consequently, InsightMapper outperforms previous methods, showcasing remarkable comprehensive performance.

## 5.2 ABLATION STUDIES

**Hybrid query generation.** Compared to naive or hierarchical query generation schemes, the proposed hybrid query generation scheme effectively blocks undesired inter-instance correlation while simultaneously maintaining inner-instance correlation. The evaluation results of the different query generation schemes are presented in Table 5. These results reveal that the hybrid query generation scheme outperforms its counterparts, thereby demonstrating the soundness of this design method.

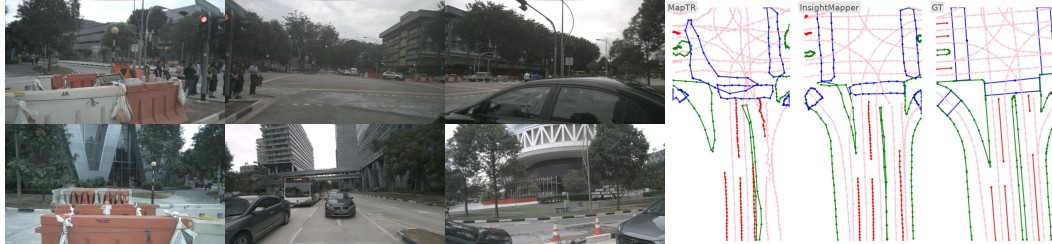

Figure 7: Qualitative visualization. The predicted map contains four classes, i.e., road boundaries (green), lane splits (red), pedestrian crossing (blue), and lane centerlines (pink).

Table 4: Efficiency of experiments. "Train" represents training time per epoch (hour). "FPS" is the inference frame rate.

| Methods | Backbone | Train | FPS | Param |
|---|---|---|---|---|
| MapTR | R18 | 0.52h | 8.5 | 35.9M |
| MapTR | R50 | 0.56h | 7.5 | 35.9M |
| InsightMapper | R18 | 0.53h | 8.5 | 45.9M |
| InsightMapper | R50 | 0.55h | 7.5 | 45.9M |

Table 5: Quantitative results of ablation studies about query generation. InsightMapper stays the same except the query generation scheme.

| Query Generation Scheme | mAP | TOPO |
|---|---|---|
| MapTR | $42.93_{(\downarrow 5.38)}$ | $46.77_{(\downarrow 4.81)}$ |
| InsightMapper-naive | $46.62_{(\downarrow 1.69)}$ | $49.39_{(\downarrow 2.19)}$ |
| InsightMapper-hierarchical | $46.93_{(\downarrow 1.38)}$ | $50.16_{(\downarrow 1.42)}$ |
| InsightMapper-hybrid | **48.31** | **51.58** |

Table 6: Quantitative results of ablation studies about query fusion.

| Query Fusion | mAP | TOPO |
|---|---|---|
| No fusion | $45.45_{(\downarrow 2.86)}$ | $48.54_{(\downarrow 3.04)}$ |
| Mean | $43.34_{(\downarrow 4.97)}$ | $46.00_{(\downarrow 5.58)}$ |
| Feed-Forward | $47.80_{(\downarrow 0.51)}$ | $50.93_{(\downarrow 0.65)}$ |
| Self-attention | **48.31** | **51.58** |

Table 7: Quantitative results of ablation studies about the masked inner-instance self-attention module.

| Method | mAP | TOPO |
|---|---|---|
| No attention | $45.42_{(\downarrow 2.89)}$ | $49.19_{(\downarrow 2.39)}$ |
| Vanilla attention | $46.23_{(\downarrow 2.08)}$ | $49.51_{(\downarrow 2.07)}$ |
| $\epsilon = 0$ | $47.67_{(\downarrow 0.64)}$ | $51.49_{(\downarrow 0.09)}$ |
| Change position | $45.24_{(\downarrow 3.07)}$ | $48.16_{(\downarrow 3.42)}$ |
| InsightMapper | **48.31** | **51.58** |

**Inner-instance query fusion.** Query fusion is designed to leverage the correlation between inner-instance queries to enhance prediction performance. Various methods can be employed for fusing queries, including mean fusing (i.e., each query is summed with the mean of all queries of the same instance), fusion by the feed-forward network (FFN), and fusion by self-attention. The evaluation results are presented in Table 6.The outcomes of the query fusion methods vary significantly, with self-attention-based query fusion achieving the best results. Consequently, self-attention query fusion is incorporated into the final design of InsightMapper.

**Inner-instance self-attention module.** In InsightMapper, the masked inner-instance self-attention module is incorporated into the decoder layers following the cross-attention module. We evaluate InsightMapper under four different conditions: without the inner-instance self-attention module (no attention), replace the inner-instance self-attention module with a vanilla self-attention module (vanilla attention), without the random blocking (no random blocking, i.e., $\epsilon=0$), and with the module placed before the cross-attention (change position). The results are displayed in Table 7. The outcomes reveal that removing the inner-instance self-attention module, removing the attention mask or altering the position of the proposed module can significantly impair model performance. This observation confirms the effectiveness of the inner-instance self-attention module design.

## 6    CONCLUSION

In this paper, we introduced InsightMapper, an end-to-end transformer-based model for on-the-fly vectorized HD map detection. InsightMapper surpasses previous works by effectively leveraging inner-instance information to improve detection outcomes. Several novel modules have been proposed to utilize inner-instance information, including hybrid query generation, inner-instance query fusion, and inner-instance feature aggregation. The superiority of InsightMapper is well demonstrated through comparative experiments on multiple datasets. InsightMapper not only enhances detection results but also maintains high efficiency, making it a promising solution for the vectorized HD map detection task.

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

## A    EXPERIMENT

### A.1    EXPERIMENT SETTINGS

In this paper, we employ a learning rate of $6 \times 10^{-4}$ and a weight decay rate of 0.01. The experiments involve 100 instance queries ($N_I = 100$) and 20 point queries ($n_p = 20$), conducted using eight NVIDIA GeForce RTX 3090 GPUs, each equipped with 24GB of memory. For the ResNet backbones, we set the batch size to 4. Our primary focus is models that utilize camera images as input, specifically six RGB camera images. But InsightMapper can be easily adapted for multi-modal perception by modifying the input of BEV transformation network.

### A.2    ROAD ELEMENTS

InsightMapper detects four types of road elements essential for vectorized HD maps: road boundaries (polyline), lane splits (polyline), pedestrian crossings (polygon), and lane centerlines (polyline with topology). Currently, the detected HD map is an undirected graph to unify all elements. If directed vectorized HD maps are required for specific applications (e.g., centerlines need directions), InsightMapper can be easily adapted by employing directed ground truth HD maps as labels to train the network. An example is shown in Table 8, where the ground truth centerlines are directed graphs.

Table 8: Quantitative results of different data representations (centerlines are directed graph).

| Method | $AP_{ped}$ | $AP_{div}$ | $AP_{bound}$ | $AP_{center}$ | mAP | TOPO |
|---|---|---|---|---|---|---|
| MapTR | 37.39 | 46.61 | 50.69 | 37.22 | 42.98 | 46.98 |
| InsightMapper | 43.24 | 53.77 | 53.89 | 43.22 | 48.53(↑5.55) | 52.09(↑5.11) |

### A.3    TOPOLOGY EVALUATION METRIC

To provide a comprehensive evaluation, we report a topology-level evaluation score in this paper, namely the TOPO metric score (He et al., 2018; 2020; He & Balakrishnan, 2022). The TOPO metric has been used in several previous studies (He et al., 2018; 2020; He & Balakrishnan, 2022) to evaluate the correctness of lane and road-network graph topology. The TOPO metric first randomly samples vertices $v_i^*$ from the ground truth graph $G^*$. It then finds corresponding matched vertices $\hat{v}_i$ in the predicted graph $\hat{G}$ based on the closest distance. Using $v_i^*$ as a seed node, TOPO calculates a sub-graph $G_{v_i^*}^*$, where the distance between all vertices and $v_i^*$ is smaller than a predetermined threshold. Similarly, we obtain the sub-graph $\hat{G}_{\hat{v}_i}$ by taking $\hat{v}_i$ as the seed node. Finally, we measure the graph similarity of the two sub-graphs using precision, recall, and F1-score. The TOPO metric is the mean similarity F1-score of all sampled vertex pairs $(v_i^*, \hat{v}_i)$.

## B    QUERY GENERATION

### B.1    DENOSING DETR (ABANDONED DESIGN)

In conventional object detection tasks, the denoising operation has been proven to effectively accelerate convergence and enhance overall performance (Li et al., 2022a; Zhang et al., 2022; Li et al., 2023a). However, this operation requires queries to be independent and identically distributed (i.i.d.). If this condition is not met, simply adding random noise to queries may not yield superior results. In our task, due to the strong correlation among inner-instance points, the denoising operation does not improve the vectorized HD map detection results. We report the outcomes of applying Dn-DETR in Table 9.

In our experiments, we initially add random instance-level noise equally to all points within the same instance. Subsequently, we introduce random point-level noise to each point. Despite these modifications, we observe neither a significant improvement nor faster convergence. We attribute this unsatisfactory performance to the correlation between points.

Table 9: Quantitative results of ablation studies about denoising DETR. For all metrics, a larger value indicates better performance.

| Method | $AP_{ped}$ | $AP_{div}$ | $AP_{bound}$ | $AP_{center}$ | mAP | TOPO |
|---|---|---|---|---|---|---|
| InsightMapper-Dn-DETR | 43.74 | 51.34 | 54.48 | 42.33 | 47.97$_{(\downarrow 0.34)}$ | 50.37$_{(\downarrow 1.21)}$ |
| InsightMapper | 44.36 | 53.36 | 52.77 | 42.35 | 48.31 | 51.58 |

## B.2 DYNAMIC QUERY GENERATION (ABANDONED DESIGN)

Another method to improve DETR in the query generation phase is dynamic query generation (Wang et al., 2022; Zhang et al., 2022; He et al., 2022). Unlike static query generation, where all queries are randomly initialized, dynamic queries are predicted based on the features obtained by the transformer encoder. In other words, the output of the transformer encoder can be utilized to generate queries with better initialization.

After obtaining the transformer encoder output $F$, we create grids to partition $F$. Each grid $F_{x,y}$ contains the local information of the input image around coordinate $(x, y)$. Then, each grid is used to predict a dynamic query, represented as a 4-D bounding box. In conventional detection tasks, objects are typically not very large, so each grid can make satisfactory predictions of dynamic queries. However, in our vectorized HD map detection task, target objects are often very thin and very long, which cannot be effectively predicted by local grids. Consequently, dynamic queries do not yield significant improvements. We present the results on dynamic query generation in Table 10.

Table 10: Quantitative results of ablation studies about dynamic queries.

| Method | $AP_{ped}$ | $AP_{div}$ | $AP_{bound}$ | $AP_{center}$ | mAP | TOPO |
|---|---|---|---|---|---|---|
| InsightMapper with dynamic query | 43.14 | 48.86 | 52.17 | 38.65 | 45.71$_{(\downarrow 2.60)}$ | 43.83$_{(\downarrow 7.75)}$ |
| InsightMapper | 44.36 | 53.36 | 52.77 | 42.35 | 48.31 | 51.58 |

## C INNER-INSTANCE FEATURE AGGREGATION

### C.1 RATIO OF MASKED ATTENTION MASK

There is an $\epsilon$ possibility that the attention mask of the inner-instance self-attention is masked (set to one, blocked). This mask operation is inspired by (He et al., 2022) to enhance the robustness of the proposed module. $\epsilon$ controls the ratio of masked attention masks. This concept is similar to dropout for better performance. However, dropout is applied after the *softmax* of attention, while the mask operation is before the *softmax*. Experimental results with different $\epsilon$ values are shown in Table 11.

Table 11: Quantitative results of ablation studies about the mask ratio.

| $\epsilon$ | $AP_{ped}$ | $AP_{div}$ | $AP_{bound}$ | $AP_{center}$ | mAP | TOPO |
|---|---|---|---|---|---|---|
| 0 (No masked attn) | 42.48 | 50.66 | 53.22 | 41.54 | 46.98$_{(\downarrow 1.33)}$ | 51.49$_{(\downarrow 0.09)}$ |
| 25% (InsightMapper) | 44.36 | 53.36 | 52.77 | 42.35 | 48.31 | 51.58 |
| 50% | 43.67 | 52.57 | 53.98 | 42.95 | 48.17$_{(\downarrow 0.14)}$ | 51.59$_{(\uparrow 0.01)}$ |
| 80% | 42.27 | 52.49 | 53.64 | 41.34 | 47.41$_{(\downarrow 0.90)}$ | 49.20$_{(\downarrow 2.38)}$ |

### C.2 TRANSFORMER ARCHITECTURES

The transformer decoder of InsightMapper may have multiple variants, which are visualized in Figure 8. The evaluation results are reported in Table 12. Based on the results, the transformer decoder has the optimal design among these variants.

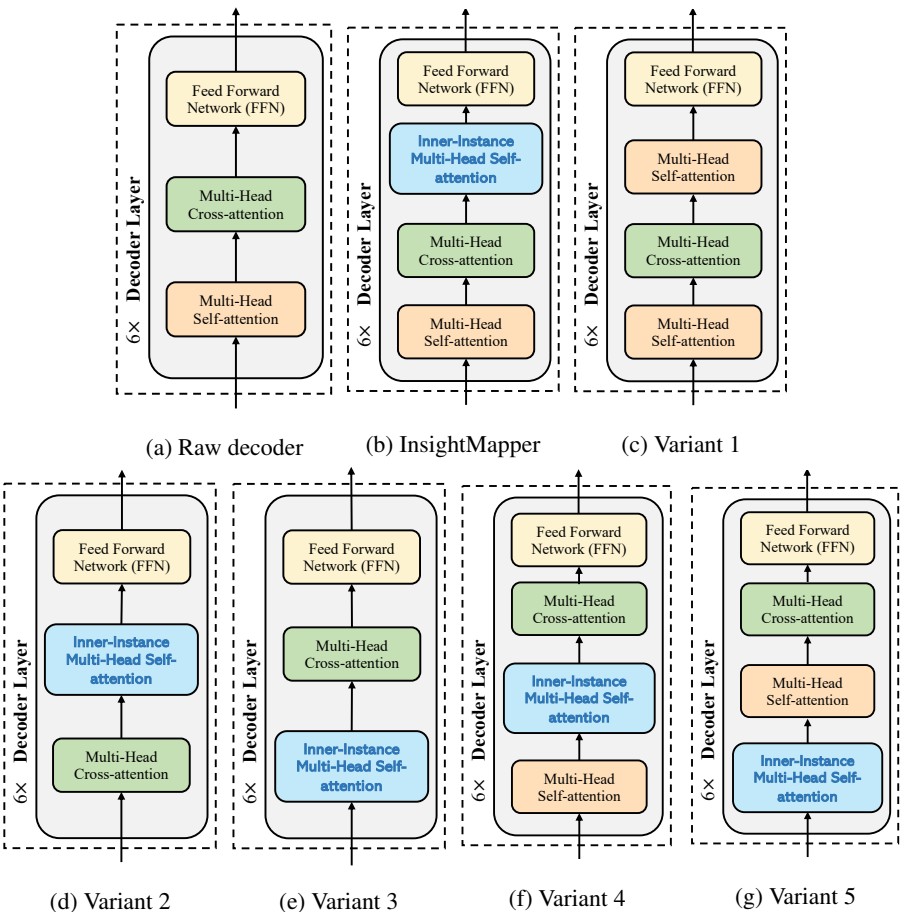

Figure 8: Different transformer decoder designs. (a) Raw decoder of the deformable DETR. (b) InsightMapper decoder. The proposed inner-instance self-attention module (blue) is incorporated. (c) Decoder variant 1. We replace the inner-instance self-attention module with a vanilla self-attention module. (d) Decoder variant 2. The self-attention module is removed from the InsightMapper decoder. (e) Decoder variant 3. Swap the inner-instance self-attention module with the cross-attention module of the variant 2. (f) Variant 4. Place the inner-instance self-attention module before the cross-attention module. (g) Variant 5. Place the inner-instance self-attention module before the self-attention module. InsightMapper decoder is the best design. Changing the structure of the proposed decoder will degrade the final performance.

Table 12: Quantitative results of ablation studies about transformer decoder architectures.

| Position | $AP_{ped}$ | $AP_{div}$ | $AP_{bound}$ | $AP_{center}$ | mAP | TOPO |
|---|---|---|---|---|---|---|
| Raw Decoder (No ins-self-attention) | 39.41 | 50.04 | 52.13 | 40.10 | $45.42_{(\downarrow 2.89)}$ | $49.19_{(\downarrow 2.39)}$ |
| Variant 1 (Normal attention) | 40.50 | 50.78 | 52.17 | 41.46 | $46.23_{(\downarrow 2.08)}$ | $49.51_{(\downarrow 1.41)}$ |
| Variant 2 | 40.96 | 47.85 | 51.11 | 40.45 | $45.09_{(\downarrow 3.22)}$ | $47.98_{(\downarrow 3.60)}$ |
| Variant 3 | 40.92 | 46.40 | 51.94 | 38.91 | $44.54_{(\downarrow 3.77)}$ | $46.90_{(\downarrow 4.68)}$ |
| Variant 4 | 39.09 | 50.09 | 51.56 | 40.21 | $45.24_{(\downarrow 3.07)}$ | $48.16_{(\downarrow 3.42)}$ |
| Variant 5 | 44.04 | 49.59 | 52.18 | 41.78 | $46.90_{(\downarrow 1.41)}$ | $49.52_{(\downarrow 2.06)}$ |
| InsightMapper | 44.36 | 53.36 | 52.77 | 42.35 | 48.31 | 51.58 |

# D  INTER-INSTANCE SELF-ATTENTION (ABANDONED DESIGN)

We also attempted to exploit the inter-instance information to further enhance the vectorized HD map detection results. The inter-instance self-attention module is incorporated into the decoder layers, similar to the inner-instance self-attention module, as shown in Figure 9.

Table 13: Quantitative results of ablation studies about inter-instance self-attention.

| Inter-instance Self-attn | $AP_{ped}$ | $AP_{div}$ | $AP_{bound}$ | $AP_{center}$ | mAP | TOPO | FPS |
|---|---|---|---|---|---|---|---|
| Yes | 42.40 | 53.21 | 54.03 | 43.03 | 48.17(↓0.14) | 52.63(↑1.05) | 6.3 |
| No (InsightMapper) | 44.36 | 53.36 | 52.77 | 42.35 | 48.31 | 51.58 | 7.7 |

First, we predict the adjacency matrix, representing the connectivity of the predicted HD map. Some lane centerline instances may intersect with each other. Adjacent instances are illustrated by colored grids. Then, for adjacent instances, the corresponding attention mask grids are set to zero (attention is allowed). Otherwise, the attention is blocked. In this way, the information exchange between points in adjacent instances is allowed to better leverage the point correlations. However, this design significantly increases resource consumption while not noticeably improving the final results. The experiment results are shown in Table 13.

We believe the reason is the sparsity of the adjacency matrix. Under most circumstances, only a few instances intersect with each other, so the adjacency matrix is very sparse, providing limited additional inter-instance information. Furthermore, it may affect the inner-instance self-attention module, which is the main reason for the performance gains of InsightMapper.

Therefore, at this stage, inter-instance self-attention is not used in InsightMapper. But this could be an interesting topic for future exploration.

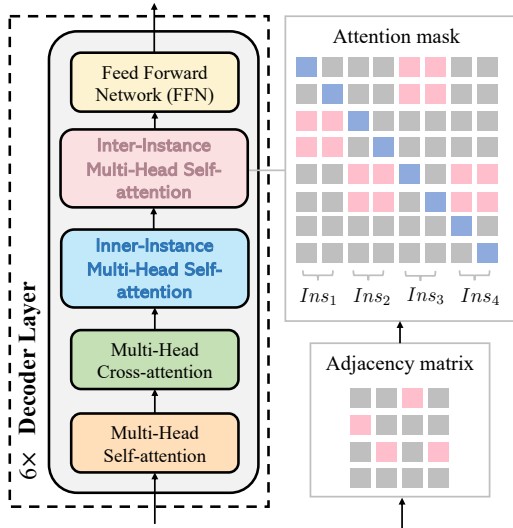

Figure 9: Decoder of InsightMapper with inter-instance self-attention module. In this module (the pink one), the attention between points of non-adjacent instances is blocked (grey grids). The attention is allowed for points of adjacent instances (pink grids), and diagonal grids (blue grids) to maintain the ego information of each point.

# E    INSTANCE-LEVEL CLASS PREDICTION

Although each instance contains $n_p$ points, it should only have one predicted class for consistency. In MapTR, after obtaining point embeddings from the transformer decoder, it calculates a new instance-level embedding by taking the mean of all points in an instance. Then, the mean embedding is sent to the class head for class prediction. Differently, in InsightMapper, we propose to use concatenation for class prediction, which preserves more information on points. Two class prediction methods are visualized in Figure 10. Experiment results are reported in Table 14. From the results, it is noted that concatenation achieves a slight performance gain. Thus, the concatenation method is used for class prediction in InsightMapper.

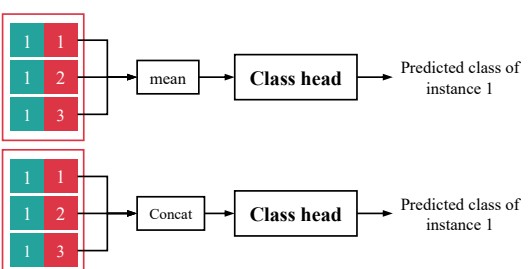

Figure 10: Class head designs. MapTR leverages the mean of points for aggregation (upper). While InsightMapper uses concatenation (lower).

Table 14: Quantitative results of ablation studies on the class head design.

| Aggregation Method | $AP_{ped}$ | $AP_{div}$ | $AP_{bound}$ | $AP_{center}$ | mAP | TOPO |
|---|---|---|---|---|---|---|
| Mean | 42.31 | 53.22 | 53.70 | 43.16 | 48.10($\downarrow$0.21) | 51.19($\downarrow$0.39) |
| Concatenation (InsightMapper) | 44.36 | 53.36 | 52.77 | 42.35 | 48.31 | 51.58 |

Table 15: Quantitative results of comparison experiments on the NuScenes *validation* set. "*" represents InsightMapper is enhanced by techniques leveraged in MapTR-V2.

| Methods | Epochs | Backbone | $AP_{ped}$ | $AP_{div}$ | $AP_{bound}$ | $AP_{center}$ | mAP | TOPO |
|---|---|---|---|---|---|---|---|---|
| MapTR-V2 | 24 | R50 | 48.20 | 54.75 | 56.66 | 45.86 | 51.37 | 51.15 |
| InsightMapper* | 24 | R50 | **49.07**($\uparrow$0.87) | **57.98**($\uparrow$3.23) | **59.84**($\uparrow$3.18) | **47.99**($\uparrow$2.13) | **53.72**($\uparrow$2.35) | **53.00**($\uparrow$1.85) |
| MapTR-V2 | 110 | R50 | 58.68 | 65.72 | 67.12 | 56.16 | 61.92 | 63.26 |
| InsightMapper* | 110 | R50 | **62.09**($\uparrow$3.41) | **67.60**($\uparrow$1.88) | **68.15**($\uparrow$1.03) | **58.41**($\uparrow$2.25) | **64.06**($\uparrow$2.14) | **64.72**($\uparrow$1.46) |

Table 16: Quantitative results of comparison experiments on the Argoverse 2 *validation* set. "*" represents InsightMapper is enhanced by techniques leveraged in MapTR-V2.

| Methods | Epochs | Backbone | $AP_{ped}$ | $AP_{div}$ | $AP_{bound}$ | $AP_{center}$ | mAP | TOPO |
|---|---|---|---|---|---|---|---|---|
| MapTR-V2 | 6 | R50 | 57.16 | 67.96 | 65.25 | 63.20 | 63.39 | 77.94 |
| InsightMapper* | 6 | R50 | **58.91**($\uparrow$1.75) | **71.35**($\uparrow$3.39) | **66.90**($\uparrow$1.65) | **64.70**($\uparrow$1.50) | **65.46**($\uparrow$2.07) | **79.26**($\uparrow$1.32) |

## F    MAPTR-V2: LATEST SOTA BASELINE

Recently (i.e., Aug/31/2023), the authors of MapTR released the implementation code of MapTR-V2 (Liao et al., 2023), an enhanced version of the original MapTR. Compared to MapTR, MapTR-V2 employs several techniques to boost the final performance and hasten convergence: 1. MapTR-V2 projects Bird's Eye View (BEV) features back to the perspective view and adds perspective-view segmentations as auxiliary supervision. 2. Drawing inspiration from (Jia et al., 2023), MapTR-V2 utilizes hybrid matching to speed up convergence and enhance the final performance.

As a result, MapTR-V2 achieves significantly improved outcomes. Nevertheless, these techniques originate from previous works and are not specifically designed for vector map detection. Therefore, we think that MapTR-V2 is optimized with regard to engineering implementation instead of theoretic improvement. To evaluate the effectiveness of our InsightMapper, we incorporate its proposed modules into MapTR-V2, further improving the detection results by more than 2%. We have made modifications to some parts of the MapTR-V2 implementation code to ensure compatibility with InsightMapper. The quantitative comparison results on the Nuscenes *validation* dataset can be found in Table 15, and the results on the Argoverse-2 dataset are shown in Table 16.

In conclusion, the three modules proposed in InsightMapper are light and flexible, and they can be easily plugged into new DETR-based frameworks to detect vector HD maps for better performance.

## G    ADDITIONAL QUALITATIVE VISUALIZATIONS

Qualitative visualizations on the Nuscenes *validation* set are shown in Figure 11 to Figure 14. The predicted map contains four classes, i.e., road boundaries (green), lane splits (red), pedestrian crossing (blue), and lane centerlines (pink). We visualize the vectorized HD map of the previous SOTA method MapTR, our proposed InsightMapper, and the ground truth label. Models are trained with ResNet50 by 24 epochs.

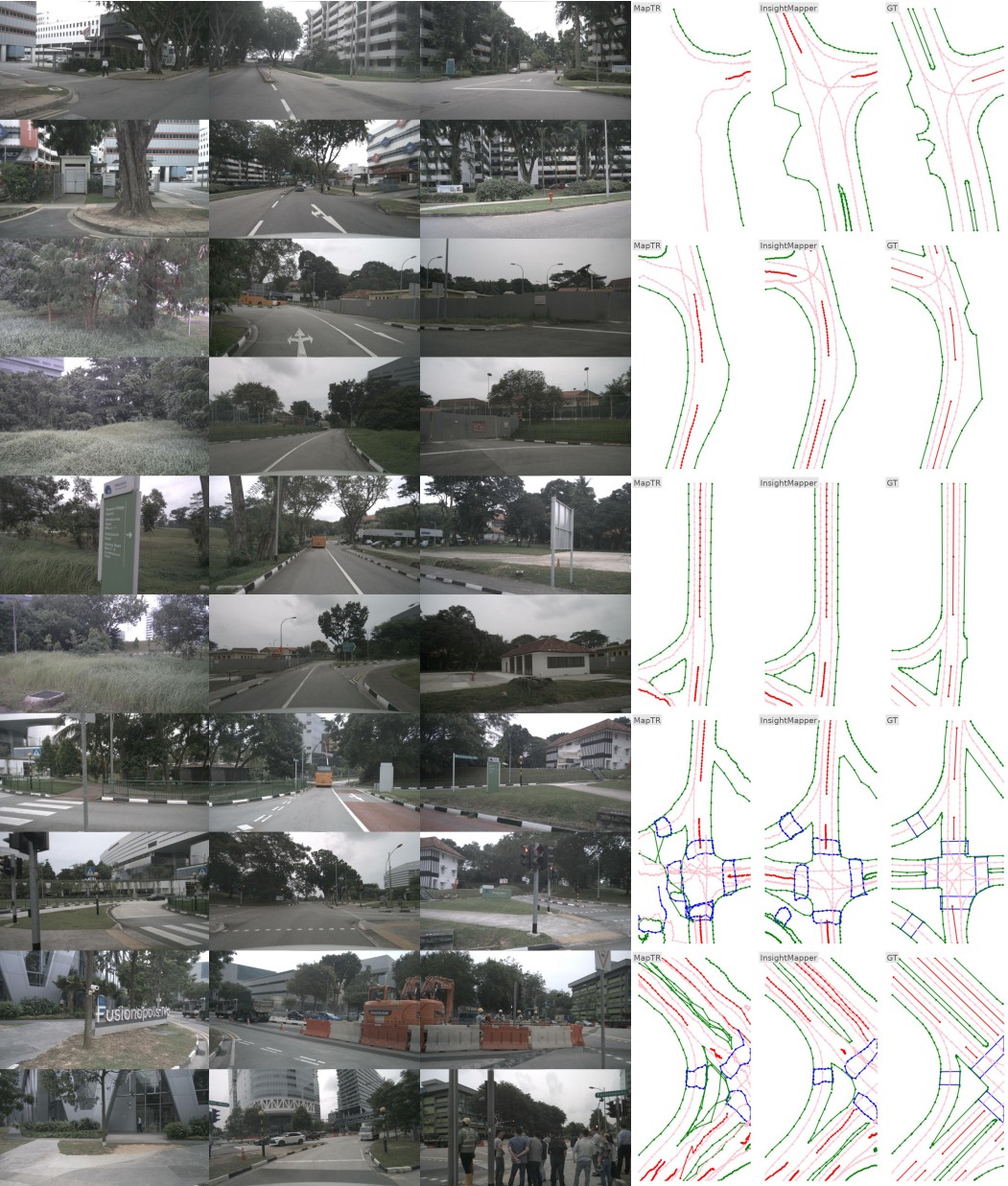

Figure 11: Qualitative visualization. Left three columns are input 6 RGB camera images. For map columns, the first column presents MapTR's results, the second column features InsightMapper's outcomes, and the final column depicts the ground truth map. The predicted map contains four classes, i.e., road boundaries (green), lane splits (red), pedestrian crossing (blue), and lane centerlines (pink).

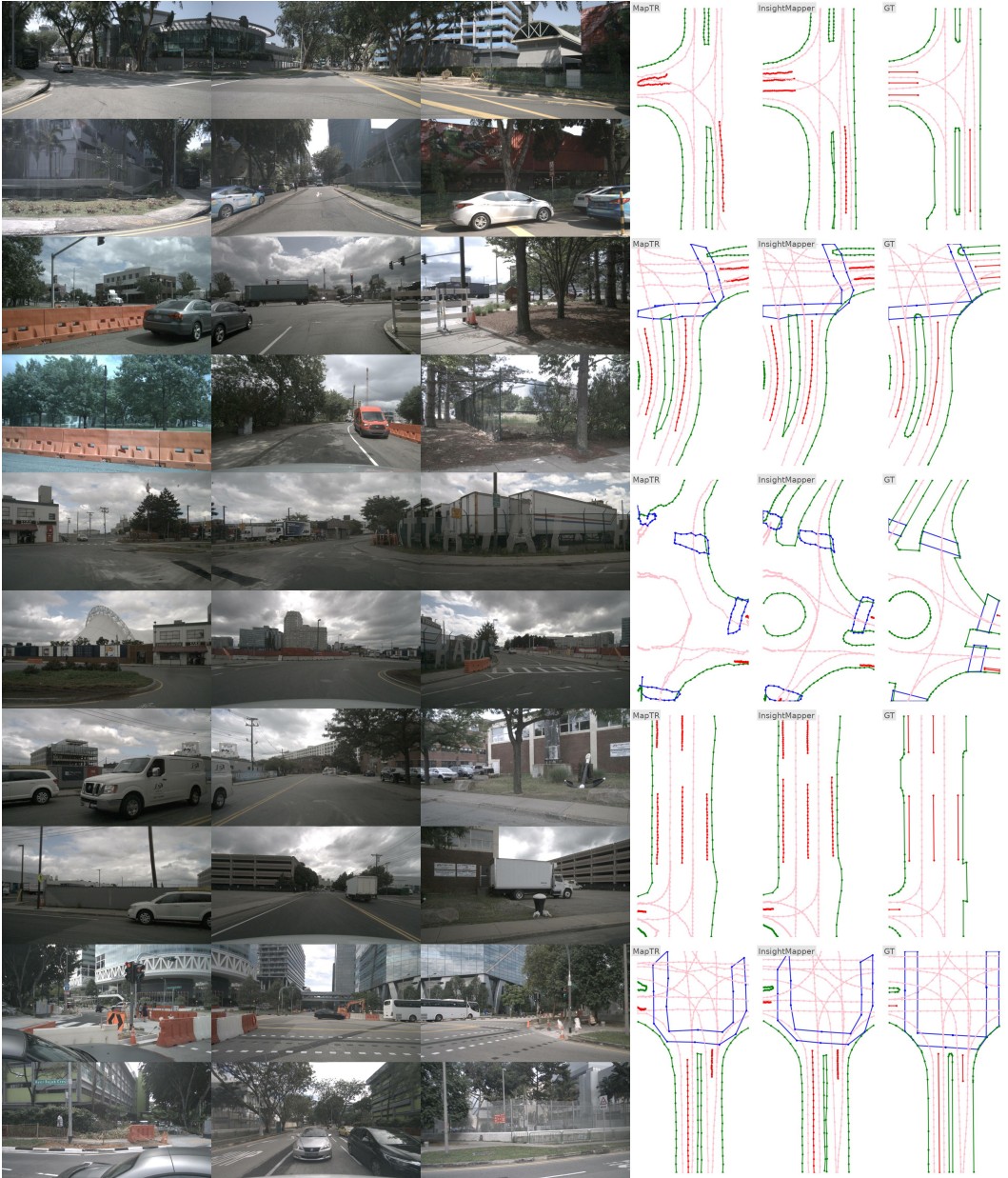

Figure 12: Qualitative visualization. Left three columns are input 6 RGB camera images. For map columns, the first column presents MapTR's results, the second column features InsightMapper's outcomes, and the final column depicts the ground truth map. The predicted map contains four classes, i.e., road boundaries (green), lane splits (red), pedestrian crossing (blue), and lane centerlines (pink).

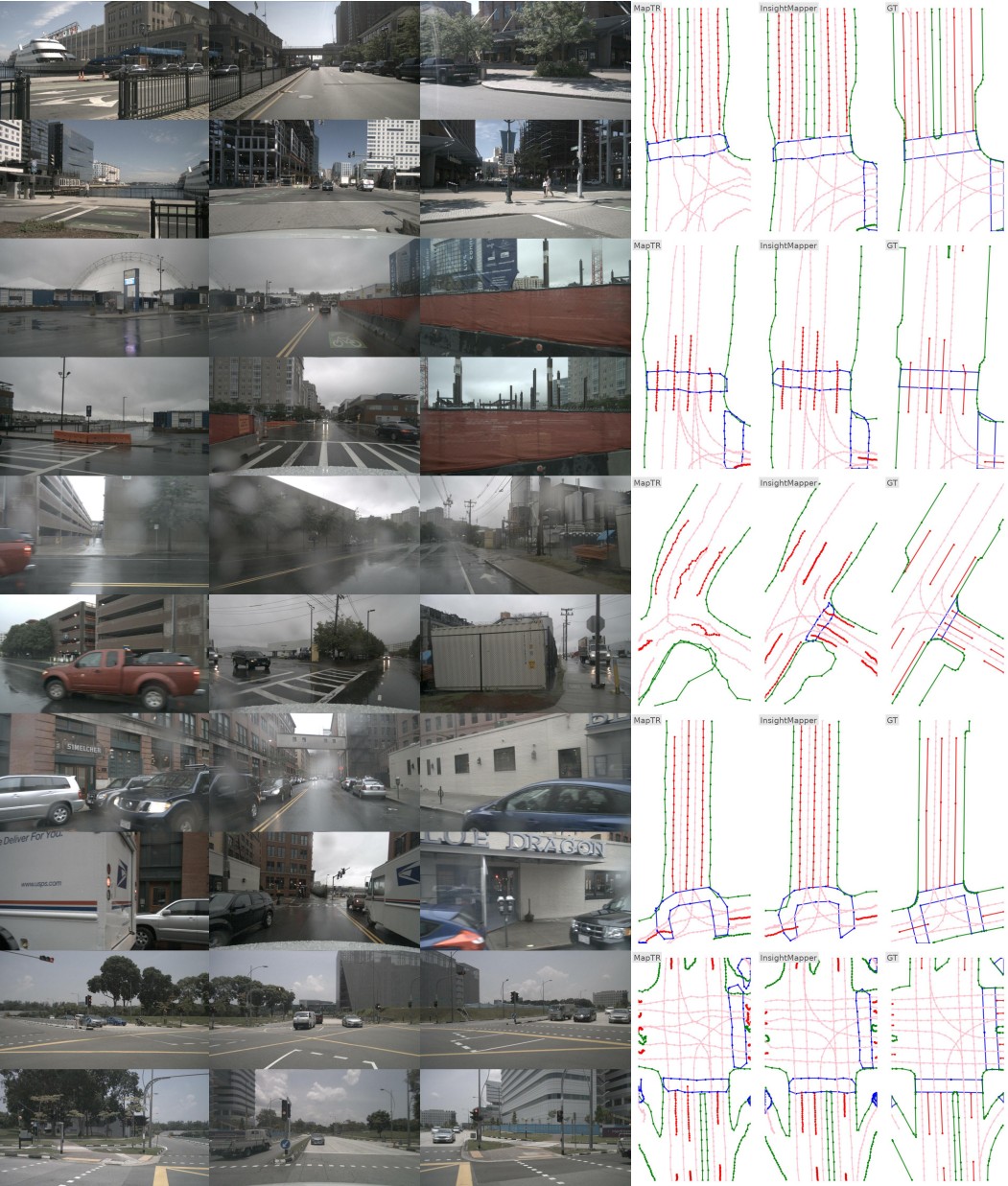

Figure 13: Qualitative visualization. Left three columns are input 6 RGB camera images. For map columns, the first column presents MapTR's results, the second column features InsightMapper's outcomes, and the final column depicts the ground truth map. The predicted map contains four classes, i.e., road boundaries (green), lane splits (red), pedestrian crossing (blue), and lane centerlines (pink).

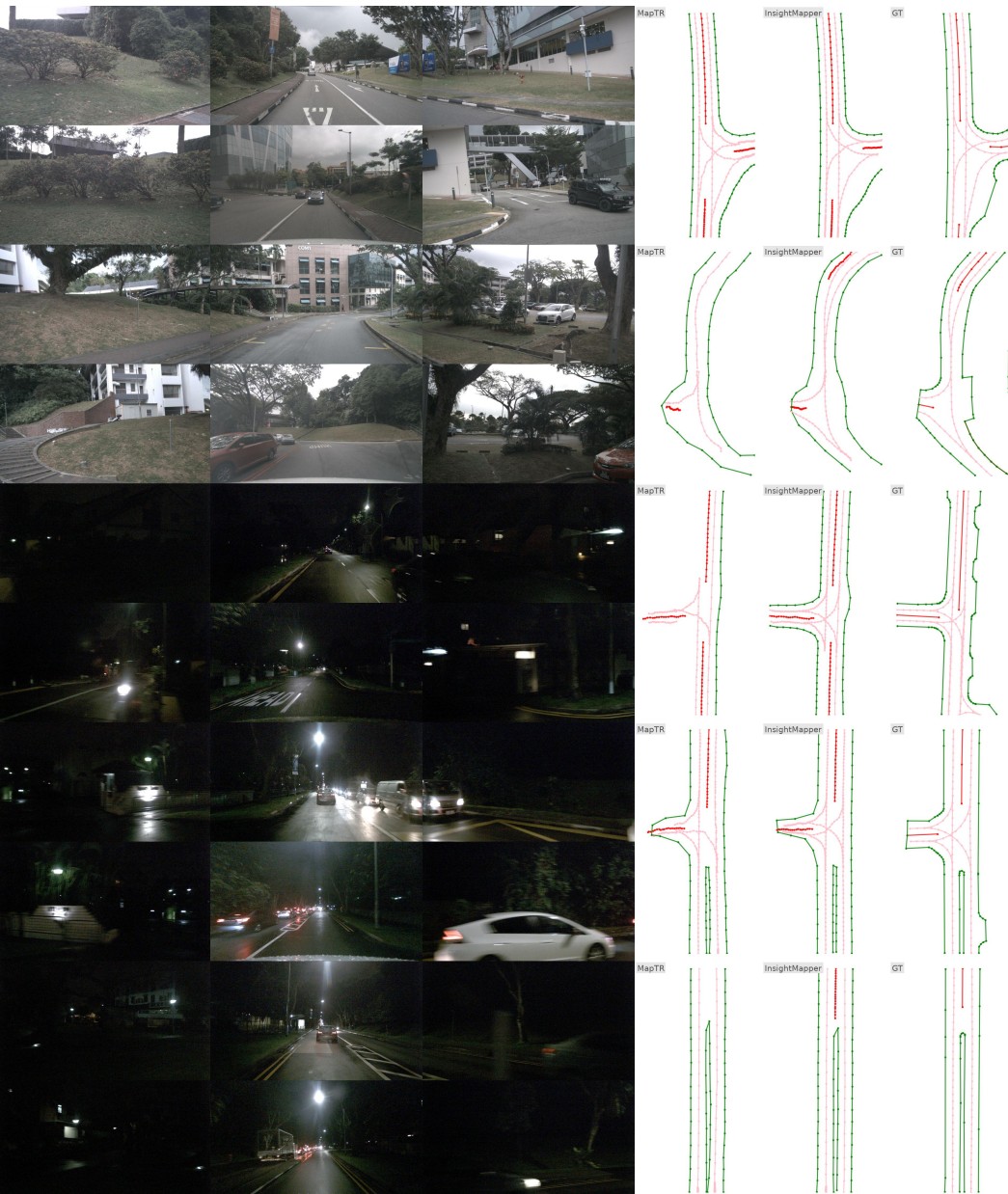

Figure 14: Qualitative visualization. Left three columns are input 6 RGB camera images. For map columns, the first column presents MapTR's results, the second column features InsightMapper's outcomes, and the final column depicts the ground truth map. The predicted map contains four classes, i.e., road boundaries (green), lane splits (red), pedestrian crossing (blue), and lane centerlines (pink).

