# OpenReview forum: "InsightMapper: A closer look at inner-instance information for vectorized High-Definition Mapping"
_ICLR.cc/2024/Conference — ICLR 2024 Conference Withdrawn Submission_

### Official Review · Reviewer_rWBZ · 2023-10-15

**Soundness:** 3 good
**Presentation:** 3 good
**Contribution:** 3 good
**Rating:** 5
**Confidence:** 4

**Summary:**

The paper presents a method to utilize inner-instance information for vectorized high-definition mapping, with the aim at improving the detection performance. The proposed method is called InsightMapper, which includes three designs to use inner-instance information. The hybrid query generation, inner-instance query fusion and inner-instance feature aggregation are proposed to construct the InsightMapper. The method is evaluated on the NuScenes and Argoverse datasets and compared to several state-of-the-art methods.

**Strengths:**

The inner-instance information and correlations are explored to improve the detection of the vectorized HD map.

In the decoder part, hybrid query is generated to maintain appropriate inner-instance information exchange as compared to the normal and hierarchical query generation.

Experimental analysis is conducted on two datasets and the results showcase the effectiveness of the proposed framework in improving the performance of the vectorized HD map detection.

**Weaknesses:**

The overall framework is constructed on top of the previous methods like BEVformer. The modification of the proposed InsightMapper framework is located in the transformer decoder for HD map detection.

The proposed hybrid query generation method sounds like an incremental version of the normal query and the hierarchical query generation methods proposed in the MapTr paper.

The ablation study of different proposed components is not included. It would be better to show the effect of the three proposed components in the whole framework for HD mapping.

**Questions:**

What is the reason for the performance drop when the self-attention applied before the cross-attention module in the inner-instance feature aggregation method?


How about the comparison with the recent HD mapping methods, such as InstaGraM [1] and Bi-Mapper [2]?

[1] Shin, J., Rameau, F., Jeong, H., & Kum, D. (2023). Instagram: Instance-level graph modeling for vectorized hd map learning. arXiv preprint arXiv:2301.04470.

[2] Li, Siyu, Kailun Yang, Hao Shi, Jiaming Zhang, Jiacheng Lin, Zhifeng Teng, and Zhiyong Li. "Bi-Mapper: Holistic BEV Semantic Mapping for Autonomous Driving." IEEE Robotics and Automation Letters 2023.


What are the performance of other previous methods on the Argoverse 2 dataset? Also, how about the performance of MapTR and InsightMapper when training with more epoch?

Apart from the FPS, how does InsightMapper compare to other methods in terms of computational complexity?

---

> ### Author Response · Authors · 2023-11-12
> **Response to reviewer rWBZ**
>
> We thank the reviewer for your detailed and insightful comments and suggestions. Here is the response and clarification to your questions and concerns.
>
>
> **1.Position of inner-instance self-attention layers**
>
> The vanilla attention used in MapTR can learn both inter-instance and inter-instance message passing. And the vanilla attention layer before the cross-attention layer (for deformable attention calculation) should treat these two kinds of message passings equally like DETR, otherwise, some instances may produce duplicate predictions since they cannot “clearly see” other instances, so that instances cannot collaborate with each other very well for global optimal prediction, which leads to severely degraded performance.
>
> The problem of the above pattern is that all queries are always equally treated, so that after the cross-attention, a point will optimize its position based on all other points with equal attention. Intuitively, points that belong to the same instance of it should project more influence. The proposed inner-instance self-attention is located after the cross-attention, and it allows a point to pay more attention to its inner-instance neighbors, to make the optimization direction of all points within an instance consistent. Otherwise,  the results might be severely affected  by the zig-zag issue.
>
> In short, the vanilla self-attention layer before the cross-attention layer assures that instances can “see” each other clearly, just like what the raw DETR does. The inner-instance self-attention layer after the cross-attention layer enables points to pay more attention to their inner-instance neighbors for local refinement. Both self-attention layers cannot be removed but they have different functions.
>
>  In the appendix, we provide detailed ablation studies in Figure. 8 and Table. 12 about attention layer position and decoder network structure. They should be helpful to further justify the design of InsightMapper.
>
> **2.Additional baseline works**
>
> Thanks for providing more related works, we will cite them in our paper. We did not consider comparing these works because: (1) InstaGram has much inferior performance than MapTR (previous SOTA), and it does not open-source the code (2) Bi-Mapper has a similar performance with HDMapNet, which is much worse than MapTR. Considering the limited page size, we mainly compare InsightMapper with past SOTA methods or most representative methods.
>
> **3.Training epoch**
>
> In Table 1, we report the results of MapTR and InsightMapper with both 24 epochs and 110 epochs training. In both epoch settings, InsightMapper greatly improves the mAP and TOPO compared with MapTR. Thus, we believe the superiority of InsightMapper is well illustrated.
>
> **4.Computation complexity**
>
> Thanks for the question. In Table 4, we add additional results for computational complexity, e.g., parameter size.
>
> **5.Ablation studies**
>
> Please refer to Table 5 to Table 7. In InsightMapper, we mainly propose three modules, and each module has multiple ablation studies to verify its effectiveness. Query generation scheme (Table. 5), query fusion (Table. 6) and inner-instance self-attention (Table. 7). We believe these ablation studies could justify the design of the proposed modules.

---

### Official Review · Reviewer_iyCD · 2023-10-27

**Soundness:** 1 poor
**Presentation:** 2 fair
**Contribution:** 2 fair
**Rating:** 3
**Confidence:** 5

**Summary:**

The paper introduces InsightMapper, a novel model for vectorized high-definition (HD) mapping, crucial for various tasks in modern autonomous vehicles such as motion planning and vehicle control. InsightMapper is designed to exploit the inner-instance correlations between predicted points, addressing the shortcomings of previous methods that failed to analyze and utilize these correlations effectively. The authors propose three novel designs within InsightMapper: Hybrid Query Generation, Inner-Instance Query Fusion and Inner-Instance Feature Aggregation. InsightMapper is evaluated using the NuScenes dataset, where it outperforms previous state-of-the-art methods by 5.78 mAP and 7.03 TOPO, showcasing its effectiveness in leveraging inner-instance information for improved performance in vectorized HD mapping. The model also maintains high efficiency during both training and inference phases, resulting in remarkable comprehensive performance.

**Strengths:**

- This work explores on the inner-instance feature interaction through mutiple modules, including inner-instance query fusion after initialized, and inner-instance point feature aggregation within the decoder layer.
- The main result is conducted on two datasets, where the performance improvement observed in InsightMapper is commendable.
- A comprehensive set of ablation experiments have been conducted to evaluate the proposed design, showcasing its effectiveness.

**Weaknesses:**

1. The authors incorporate centerline perception into their task as a novel addition. During preprocessing, they deliberately omit centerlines within intersections that have degrees greater than two, aiming to simplify the learning of lane topology. However, this modification compromises the integrity and consistency of the topology, rendering the centerline results somewhat inconsequential. The visual representation of both the ground truth and prediction appears weird and lacks clarity.

2. The authors assert that "Usually, inter-instance correlation distracts the inner-instance information exchange, degrading the final performance". They also maintain that "self-attention before cross-attention should treat all queries equally to prevent duplicated predictions". However, there seems to be an inconsistency in these claims. The self-attention in decoder layers can also foster inter-instrance correlation, which is necessary for DETR-like detection paradigm. The assertions appear somewhat unsubstantiated, particularly when considering that eliminating the inter-instance mask in the inner-instance self-attention module results in a marginal decrease of -0.6% in mAP and -0.09% in TOPO. The notable contribution of the module, reflected in a +2.89% increase in mAP and 2.39 in TOPO, seems to be primarily due to the introduction of an additional attention layer and its strategic positioning, rather than the mitigation of distractions caused by inter-instance correlation.

3. The paper falls short in providing comprehensive implementation details, both regarding the proposed method and the re-training processes of other state-of-the-art (SOTA) methods. Critical information such as the input image resolution and the number of layers in both the encoder and decoder is omitted. This lack of transparency raises questions about the fairness and comparability among the evaluated methods.

4. The paper's novelty and contributions are moderate. The paper demonstrates a commendable performance gain, but it appears that this improvement predominantly stems from the addition of layers. While it presents new modules and findings, the mechanisms driving this enhancement are not elucidated clearly. The theoretical foundation and analysis presented in the paper seem to lack depth and solidity.

**Questions:**

1. In reference to the weakness 1, what is the essential rationale behind the modification? While it is noted that connectivity with intersections is not entirely eliminated, the remaining components appear to lack substantive meaning or relevance. Could you clarify the necessity of this adjustment in the context of your study’s objectives and outcomes?

2. Can you report the result of your model on the more popular benchmark with only 3 classes? So you can perform a fair comparision with the official results of MapTR, VectorMapNet, PivotNet etc..

3. In reference to the weakness 2, could you elucidate how the distraction caused by inter-instance correlation contributes to the degradation of the final performance? From my perspective, the effectiveness of the additional self-attention layer seems to originate from the enhancement of inner-instance feature aggregation rather than the blocking of inter-instance correlation (because the inter-instance correlation in self-attention layer is still exist). Could you provide further clarification on this aspect?

4. Could you clarify what is the difference between "Vanilla attention" and "No mask" in the Table 7? It seems the only difference that you have mentioned is the randomly mask in your inner-instance points. The gap between this two models seems huge (1.4 on mAP). Can you give a further explanation?

5. In reference to the weakness 3, could you furnish the more detailed information regarding the implementation? Considering the relatively low FPS reported, it appears that substantial modifications have been made to the original design of MapTR, as well as to other methods employed in the study.

6. Regarding different query generation schemes, can you specify which research utilizes the naive query generation pipeline? It has come to my attention that a recent work, PivotNet [1], seems to utilize a strategy similar to your query generation approach. Could you also provide comparison between the PivotNet?

7. Some minor issues:

   - In your third paragraph of introduction, PolyDiffuse (Chen et al., 2023) and TopoNet (Li et al., 2023b) are not segmentation-based mothed.

   - missing period in end of Section 3.1.

   - wrong quotation mark, ”blind.” in Section 4.3.

---

> ### Author Response · Authors · 2023-11-12
> **Response to reviewer iyCD (part1/2)**
>
> We thank the reviewer for your detailed and insightful comments and suggestions. Here is the response and clarification to your questions and concerns.
>
> **1.Data preprocessing**
>
> Thanks for the question. At this stage, to make centerline predictable by DETR-based models like other road elements, it has to be represented by polylines instead of complicated graph with intersections. This data preprocessing method has been widely used in past works that focused on centerline detection. Directly detecting centerlines by DETR-based models without decompensating them into polylines could be treated as promising future work.
>
> **2.Evaluation with 3 classes**
> For better comparison with MapTR, we report the evaluation results with only 3 classes (i.e., boundary, ped walking and road divide). You may find the results in Table 4 in the revised manuscript. Compared with 4-class road element detection, 3-class detection is much simpler, so the mAP improvement of 3-class detection is as high as 4-class detection, but it still reaches around +3mAP. Similar to the evaluation of Argoverse-2 in Table 3, the more complex the detection task/scenario is, the better performance gain can InsightMapper achieve.
>
> In the comparison experiments, we keep all experiment settings exactly the same for MapTR and InsightMapper, except proposed plugged-in modules.
> | Method      | Epochs      | Backbone     | Modality      | $AP_{ped}$      | $AP_{div}$      | $AP_{bound}$      | mAP      |
> | ----------- | ----------- | ----------- | ----------- | ----------- | ----------- | ----------- | ----------- |
> | MapTR      | 24       | R50      | C       | 46.04      | 51.58       | 53.08      | 50.23       |
> | InsightMapper   | 24        | R50      | C       | 48.44      | 54.68       | 56.92      | 53.35       |
>
>
> **3.Inter-instance and inner-instance**
>
> We apologize for inappropriate definitions and insufficient explanations. You might misunderstand the design of the masked inner-instance self-attention module. Similar analysis can also be found in answer 4. “No mask” does not mean removing the mask blocking inter-instance information, it indicates we do not block inner-instance attention with probability $\epsilon$ for better robustness, inspired by Masked AutoEncoder. We will change “no mask” into "no random blocking($\epsilon=0$)" in the revised manuscript. It should noted that inter-instance information exchange is always blocked in the inner-instance self-attention module.
>
> In your review, there are two claims thought to be contradictory: claim A."Usually, inter-instance correlation distracts the inner-instance information exchange, degrading the final performance", and claim B. "self-attention before cross-attention should treat all queries equally to prevent duplicated predictions". In fact, these two claims are not contradictory since they have different purposes.
>
> In claim A, we talked about how inner-instance and inter-instance information exchange affect the line refinement and point position optimization. This information exchange happens after the cross-attention. Intuitively, points that belong to the same instance of it should project more influence. The proposed inner-instance self-attention allows a point to pay more attention to its inner-instance neighbors, to make the optimization direction of all points within an instance consistent. Otherwise,  the results might be severely affected  by the zig-zag issue.
>
> In claim B, we talked about how self-attention works for deformable cross-attention. This happens before the cross-attention. The self-attention layer should treat inter- and inner-message passings equally, otherwise, some instances may produce duplicate predictions since they cannot “clearly see” other instances, so instances cannot collaborate with each other very well for global optimal prediction, which leads to severely degraded performance.
>
> Thus, claim A and claim B do not contradict each other, and they are claims for different purposes. Violating either claim may lead to degraded results.

---

> ### Author Response · Authors · 2023-11-12
> **Response to reviewer iyCD**
>
> **4.Vanilla attention and No mask**
>
> Sorry for the inappropriate definition and insufficient explanation. Vanilla attention indicates attention layers used in MapTR, which treats inner-instance and inter-instance information equally. “No mask” indicates that the inner-instance self-attention layer does not block inner-instance self-attention with probability $\epsilon$. In section 4.4 and Figure.6, we proposed to randomly block some attentions between inner-instance points with probability $\epsilon$ for better robustness, inspired by Masked Autoencoder. The expression “No mask” could be misleading, and we changed it to “No random blocking($\epsilon=0$)” in the revised manuscript. You may find the results with different $\epsilon$ in Table. 11 of the appendix.
>
>
>
>
> **5.Implementation settings**
>
> In our experiments, we strictly followed MapTR for parameter settings. Except the parameters reported in the experiment results (Table 1, epochs, backbone, modality), we keep all other settings exactly the same for fair comparison, e.g., batch size, input data resolution, network structure (e.g., layers of decoder), input query numbers, embedding dimension, etc.  Thus, the fairness of the experiments is assured.
>
> **6. FPS and network structure**
>
> The FPS reported in the raw MapTR (Res50) is about 15. The FPS reported in InsightMapper is 7.5. This is because we detect centerlines as an additional class. The number of centerline classes is much larger than other classes, which makes the frame rate drop. But we also provide the FPS of MapTR for 4 classes detection (8.5FPS), which is very close to the FPS of InsightMapper. This indicates that InsightMapper is very efficient, considering it greatly improves mAP and TOPO but minorly affects the FPS.
>
> The drop of FPS comes from additional inner-instance layers, which is minor, and we did not modify other parts of MapTR, like encoder, decoder layers, input query numbers, input image resolutions, etc.
>
> **7.Query generation schemes**
>
> Strictly speaking, it seems that there is no work for HD map detection utilizing the naïve scheme. But DETR/deformable DETR for object detection widely adopts such query generation scheme, and that’s why we add a comparison with the naïve scheme. PivotNet has variant input query length, which makes their query generation problems different from ours. Since PivotNet is a quite newly released paper and it does not open-source its code, we are afraid we cannot compare with it at this stage. But corresponding comparison experiments could be conducted in the future as long as the authors release their code. In that way, probably the proposed modules in InsightMapper could be plugged into PivotNet for further improvement.
>
> **8.Contributions**
>
> InsightMapper proposed simple but effective modules for better HD map detection. Although these ideas might not be complicated,  corresponding insights about utilizing inner-instance information and improved results still make contributions to the community.

---

### Official Review · Reviewer_658H · 2023-10-30

**Soundness:** 2 fair
**Presentation:** 3 good
**Contribution:** 2 fair
**Rating:** 5
**Confidence:** 4

**Summary:**

This paper studies the problem of vectorized high-definition mapping, which aims to reconstruct vectorized maps from onboard sensors (e.g., cameras, LiDARs). Building upon a previous Transformer-based method, MapTR, the authors study the attention designs of the Transformer decoder and propose three improvements based on empirical findings: i) assigning each point a unique point-level query embedding, ii) adding an additional self-attention layer before the Transformer decoder,  iii) adding additional self-attention layers inside the Transformer decoder, dedicated for intra-instance modeling. With these modifications, the proposed method, InsightMapper, improves MapTR on both nuScenes and Argoverse datasets. Extensive quantitative and qualitative results are provided.

**Strengths:**

- The paper provides a set of reasonable explorations on the query and attention designs for the HD mapping task. Since similar Transformer-based or DETR-based frameworks are general and can be applied to many different computer vision problems (e.g., geometry generation/reconstruction), the findings in this paper can also be helpful in other domains.

- The experiments are extensive and show good quantitative performance on both the nuScenes and Argoverse datasets. The supplementary doc also covers recent works like MapTR-v2, providing strong empirical results.

- The supplementary video provides good qualitative comparisons;

**Weaknesses:**

This paper presents promising empirical results. However, my concern is that the method design lacks convincing explanations and insights, both theoretically and empirically. The reasonings in Sec.3 and Sec.4 are mostly subjective conjectures drawn from the experimental trials and errors, which could not accurately explain the failure of previous designs. Details are listed below:

(1). Ideally, with the hierarchical query design in MapTR, the self-attention layers of the Transformer decoder can also learn inter-instance and inner-instance message passing. This paper does not convincingly explain why the existing designs failed. Words alone are not persuasive enough, and I expect more detailed/principled analyses (e.g., showing the attention patterns between the points learned by MapTR and the proposed method).

(2). Recent works like MapTR-v2 decompose the global self-attention layer into inter-instance and intra-instance self-attention, which reduces the memory and computational cost but *does not significantly improve the mAP*. This again makes the paper's reasonings/explanations unconvincing because MapTR-v2's intra-instance attention looks very similar to the proposed inner-instance self-attention -- is the difference in the order of the attention layers inside each decoder block?

(3). I can not buy the arguments for the "hybrid query generation" and "hierarchical query generation" unless more in-depth analyses like the attention weights visualization are provided. Sharing the point-level query across instances is natural as the point embedding can encode the spatial affinity of points; for example, $q_2^P$ is spatially closer to $q_1^P$ than $q_{10}^P$, which applies to most polylines. The "hybrid query generation" and the "inner-instance query fusion" add additional modeling capacity and improve the quantitative results, but I cannot see clear insights on why these changes can make such huge differences.


Without convincing explanations, the contributions of this paper are limited to empirical trials and errors, and it may not further inspire future works in this area.

**Questions:**

Besides the concerns discussed in the weaknesses section, there are two minor questions:

- I appreciate that the evaluation protocol used in this paper considers the additional "centerline" class, which is more challenging than the conventional evaluation setting in previous works (e.g., VectorMapNet, MapTR, etc.). However, it is helpful to also provide the results on the old benchmark without the "centerline" class. There are two reasons: i) it can show if the proposed method works better when the tasks become more challenging by comparing the performance gap with and without centerlines; ii) numbers from previous papers can be directly compared in a table, making the results more consistent across different papers.

- In Table 7, the difference between "Vanilla attention" and "No mask" is unclear. From the descriptions in Sec.4.4, it seems that the "attention without the mask" is equivalent to the "vanilla attention"?

---

> ### Author Response · Authors · 2023-11-12
> **Response to reviewer 658H (part 1/2)**
>
> We thank the reviewer for your detailed and insightful comments and suggestions. Here is the response and clarification to your questions and concerns.
>
> **1.Attention patterns of MapTR and InsightMapper**
>
> The vanilla attention used in MapTR can learn both inter-instance and inter-instance message passing. The vanilla attention layer before the cross-attention layer (for deformable attention calculation) should treat these two kinds of message passings equally like DETR, otherwise, some instances may produce duplicate predictions since they cannot “clearly see” other instances so that instances cannot collaborate with each other very well for global optimal prediction, which leads to severely degraded performance.
>
> The problem with the above pattern is that all queries are always equally treated so that after the cross-attention, a point will optimize its position based on all other points with equal attention. Intuitively, points that belong to the same instance of it should project more influence. The proposed inner-instance self-attention is located after the cross-attention, and it allows a point to pay more attention to its inner-instance neighbors, to make the optimization direction of all points within an instance consistent. Otherwise,  the results might be severely affected by the zig-zag issue.
>
> In short, the vanilla self-attention layer before the cross-attention layer assures that instances can “see” each other clearly, just like what the raw DETR does. The inner-instance self-attention layer after the cross-attention layer enables points to pay more attention to their inner-instance neighbors for local refinement. Both self-attention layers cannot be removed but they have different functions.
>
>  In the appendix, we provide detailed ablation studies in Figure. 8 and Table. 12 about attention layer position and decoder network structure. They should be helpful to further justify the design of InsightMapper.
>
> **2.Difference with MapTR-V2**
>
> In answer 1, we analyzed the function and properties of two kinds self-attention layers before (I.e., vanilla self-attention layer) and after (I.e., proposed inner-instance self-attention layer) the cross-attention layer. Both kinds of self-attention layers are essential for expected results. Therefore, even though MapTR-V2 decomposes the vanilla self-attention layer into inter self-attention and intra-self-attention, it only works on the self-attention layer before the cross-attention layer, and it fails to aggregate inner-instance information after the cross-attention layer for line refinement. Queries in MapTR-V2 still cast equal attention to all points, instead of paying more attention to inner-instance neighbors.
>
> **3.Hybrid query generation and hierarchical query generation**
>
> Thanks for the question. Indeed $q_2^P$ is spatially closer to $q_1^P$ than $q_{10}^P$. But the hierarchical query scheme introduces unexpected correlations between the i-th points of different instances. For example, $q_{1,1}$ indicates the 1-st point in the 1-st instance, and $q_{10,1}$ indicates the 1-st point in the 10-th instance. These two queries should not have correlations at the very beginning, otherwise the optimization of $q_{1,1}$ might affect that of $q_{10,1}$. The unexpected correlation cast unwanted constraints on the model optimization, which leads to degraded performance. In this way, the i-th points of all instances have such unwanted correlations (i.e., $\{q_{1,i},q_{2,i},...,q_{N,i}\}$) and this unwanted constraints should be blocked. In InsightMapper, the proposed hybrid query generation blocks such correlations (points in different instances are independent) and better performance is achieved.

---

> ### Author Response · Authors · 2023-11-12
> **Response to reviewer 658H (part 2/2)**
>
> **4.Evalution with 3 classes**
> For better comparison with MapTR, we report the evaluation results with only 3 classes (i.e., boundary, ped walking and road divide). You may find the results in Table 4 in the revised manuscript. Compared with 4-class road element detection, 3-class detection is much simpler, so the mAP improvement of 3-class detection is as high as 4-class detection, but it still reaches around +3mAP. Similar to the evaluation of Argoverse-2 in Table 3, the more complex the detection task/scenario is, the better performance gain can InsightMapper achieve.
>
> In the comparison experiments, we keep all experiment settings exactly the same for MapTR and InsightMapper, except proposed plugged-in modules.
> | Method      | Epochs      | Backbone     | Modality      | $AP_{ped}$      | $AP_{div}$      | $AP_{bound}$      | mAP      |
> | ----------- | ----------- | ----------- | ----------- | ----------- | ----------- | ----------- | ----------- |
> | MapTR      | 24       | R50      | C       | 46.04      | 51.58       | 53.08      | 50.23       |
> | InsightMapper   | 24        | R50      | C       | 48.44      | 54.68       | 56.92      | 53.35       |
>
>
> **5.Vanilla attention and No mask**
>
> Sorry for the inappropriate definition and insufficient explanation. Vanilla attention indicates attention layers used in MapTR, which treats inner-instance and inter-instance information equally. “No mask” indicates that the inner-instance self-attention layer does not block inner-instance self-attention with probability $\epsilon$. In section 4.4 and Figure.6, we proposed to randomly block some attentions between inner-instance points with probability $\epsilon$ for better robustness, inspired by Masked Autoencoder. The expression “No mask” could be misleading, and we changed it to “No random blocking($\epsilon=0$)” in the revised manuscript. You may find the results with different $\epsilon$ in Table. 11 of the appendix.

---

### Official Review · Reviewer_wpLq · 2023-10-30

**Soundness:** 1 poor
**Presentation:** 2 fair
**Contribution:** 2 fair
**Rating:** 5
**Confidence:** 3

**Summary:**

In this paper, the authors proposed InsightMapper, which predicts online vectorized HD maps from multiview images. Compared with prior works, the authors proposed to leverage inner-instance information when predicting map objects: 1) hybrid query generation that consists of both instance-level and point (sub-instance) level queries without reusing the point-level queries 2) inner-instance query fusion that computes better point-level queries by weighted sum 3) inner-instance feature aggregation layer that limits the attention within each instance. The authors showed that the proposed method can yield better performance than the previous SOTA (MapTR) on both nuScenes and Argoverse2 datasets.

**Strengths:**

- The paper is overall well written and easy to follow.
- The introduction and the related works sections summarized quite a complete set of recent HD map detection works.
- The key idea is simple and it makes sense. The authors do a good job of ablating the model to support the three designs.
- The proposed method achieves SOTA results on nuScenes and Argoverse2 datasets.

**Weaknesses:**

1. Numbers do not match those in MapTR. I am a bit confused as the numbers reported in Table 1 for the MapTR method seem not to match those reported in the original paper (https://openreview.net/pdf?id=k7p_YAO7yE). E.g. for MapTR 24 epoch R50 model, the mAP listed in Table 1 is 42.93, while in the original paper, the number is 50.3 (see Table 1 of the MapTR paper). I am not sure if I missed anything (is the training/evaluation setting different?), but I did not find anywhere in the paper explaining this difference.

2. Questions on inner-instance query fusion: in equation (3), the $w_{i,j,k}$ is not defined. How do you compute or obtain the weights? Are they just learnable weights?

3. Question about Table 7. What is the difference between "Vanilla attention" and "No mask"? If I understand correctly, the proposed inner-instance self-attention module is vanilla attention with a mask, and in this case, "Vanilla attention" and "No mask" should refer to the same module.

Minor:
- In the 11th row of the **HD Map Detection** paragraph of the related works section, the VectrorMapNet citation is wrong (it pointed to VectorNet).

**Questions:**

I am mostly concerned about the first issue in the weaknesses section, the improvement is no longer this significant compared with the original numbers from MapTR. And I am also concerned about the Q2 and Q3 for the clarity of the writing.

I have not directly worked on HDMap detection before, thus my judgment on the SOTA results and method novelty might not be accurate.

---

> ### Author Response · Authors · 2023-11-12
> **Response to reviewer wpLq**
>
> We thank the reviewer for your detailed and insightful comments and suggestions. Here is the response and clarification to your questions and concerns.
>
> **1.Evaluation results compared with the original numbers from MapTR**
>
> In the original paper of MapTR, the authors only consider 3 classes of road elements (i.e., boundary, road split, and pedestrian walking). In InsightMapper, we add another new class “lane centerline” into the prediction, which enables our model to handle more road elements. Therefore, it is not appropriate to directly compare the results reported in InsightMapper (4 class prediction) and those in MapTR (3 class prediction).
>
> **2.Notation of inner-instance query fusion**
>
> Sorry for the lack of definition of notation. $w_{i,j,k}$ represent weights for query fusion, which could be learnable weights of fully connected layers or self-attention layers.
>
> **3.Vanilla attention and “No mask”**
>
> Sorry for the inappropriate definition and insufficient explanation. Vanilla attention indicates attention layers used in MapTR, which treats inner-instance and inter-instance information equally. “No mask” indicates the inner-instance self-attention layer does not block inner-instance self-attention with probability $\epsilon$. In section 4.4 and Figure.6, we proposed to randomly block some attentions between inner-instance points with probability $\epsilon$ for better robustness, inspried by Masked Autoencoder. The expression “No mask” could be misleading, and we change it to “No random blocking($\epsilon=0$)” in the revised manuscript. You may find the results with different $\epsilon$ in Table. 11 of the appendix.

---

### Author Response · Authors · 2023-11-12
**Response to reviewers**

We thank all reviewers for your comments and detailed feedback on this paper! We uploaded the version-1 revised manuscript, and we highlighted changes with blue text. Thanks again for your efforts.

Here are some explanations and details about the concepts in our paper and the revisions we have just made. Since multiple reviewers have questions about them, we make a declaration here for your reference.

**1.Vanilla attention and "No mask"**

Sorry for the inappropriate definition and insufficient explanation. Vanilla attention indicates attention layers used in MapTR, which treats inner-instance and inter-instance information equally. “No mask” indicates that the inner-instance self-attention layer does not block inner-instance self-attention with probability $\epsilon$. In section 4.4 and Figure.6, we proposed to randomly block some attentions between inner-instance points with probability $\epsilon$ for better robustness, inspired by Masked Autoencoder. The expression “No mask” could be misleading, and we changed it to “No random blocking($\epsilon=0$)” in the revised manuscript. You may find the results with different $\epsilon$ in Table. 11 of the appendix.

**2.Evaluation results with 3 classes**

For better comparison with MapTR, we report the evaluation results with only 3 classes (i.e., boundary, ped walking and road divide). You may find the results in Table 4 in the revised manuscript. Compared with 4-class road element detection, 3-class detection is much simpler, so the mAP improvement of 3-class detection is as high as 4-class detection, but it still reaches around +3mAP. Similar to the evaluation of Argoverse-2 in Table 3, the more complex the detection task/scenario is, the better performance gain can InsightMapper achieve.

In the comparison experiments, we keep all experiment settings exactly the same for MapTR and InsightMapper, except proposed plugged-in modules.
| Method      | Epochs      | Backbone     | Modality      | $AP_{ped}$      | $AP_{div}$      | $AP_{bound}$      | mAP      |
| ----------- | ----------- | ----------- | ----------- | ----------- | ----------- | ----------- | ----------- |
| MapTR      | 24       | R50      | C       | 46.04      | 51.58       | 53.08      | 50.23       |
| InsightMapper   | 24        | R50      | C       | 48.44      | 54.68       | 56.92      | 53.35       |

**3.Self-attention layers before and after the cross-attention layer**

The vanilla attention used in MapTR can learn both inter-instance and inter-instance message passing. And the vanilla attention layer before the cross-attention layer (for deformable attention calculation) should treat these two kinds of message passings equally like DETR, otherwise, some instances may produce duplicate predictions since they cannot “clearly see” other instances, so that instances cannot collaborate with each other very well for global optimal prediction, which leads to severely degraded performance.

The problem of the above pattern is that all queries are always equally treated so that after the cross-attention, a point will optimize its position based on all other points with equal attention. Intuitively, points that belong to the same instance of it should project more influence. The proposed inner-instance self-attention is located after the cross-attention, and it allows a point to pay more attention to its inner-instance neighbors, to make the optimization direction of all points within an instance consistent. Otherwise,  the results might be severely affected  by the zig-zag issue.

In short, the vanilla self-attention layer before the cross-attention layer assures that instances can “see” each other clearly, just like what the raw DETR does. The inner-instance self-attention layer after the cross-attention layer enables points to pay more attention to their inner-instance neighbors for local refinement. Both self-attention layers cannot be removed but they have different functions.

In the appendix, we provide detailed ablation studies in Figure. 8 and Table. 12 about attention layer position and decoder network structure. They should be helpful to further justify the design of InsightMapper.